# Machine learning and deep learning techniques to support clinical diagnosis of arboviral diseases: A systematic review

**Sebastião Rogério da Silva Neto**[1], **Thomás Tabosa Oliveira**[1], **Igor Vitor Teixeira**[1], **Samuel Benjamin Aguiar de Oliveira**[2,3], **Vanderson Souza Sampaio**[2,3], **Theo Lynn**[4], **Patricia Takako Endo**[1]*

**1** Universidade de Pernambuco, Recife, Brazil, **2** Universidade do Estado do Amazonas, Manaus, Brazil, **3** Fundação de Medicina Tropical Dr. Heitor Vieira Dourado, Manaus, Brazil, **4** Dublin City University, Dublin, Ireland

* patricia.endo@upe.br

**Data Availability Statement:** All relevant data are within the manuscript.

## Abstract

### Background

Neglected tropical diseases (NTDs) primarily affect the poorest populations, often living in remote, rural areas, urban slums or conflict zones. Arboviruses are a significant NTD category spread by mosquitoes. Dengue, Chikungunya, and Zika are three arboviruses that affect a large proportion of the population in Latin and South America. The clinical diagnosis of these arboviral diseases is a difficult task due to the concurrent circulation of several arboviruses which present similar symptoms, inaccurate serologic tests resulting from cross-reaction and co-infection with other arboviruses.

### Objective

The goal of this paper is to present evidence on the state of the art of studies investigating the automatic classification of arboviral diseases to support clinical diagnosis based on Machine Learning (ML) and Deep Learning (DL) models.

### Method

We carried out a Systematic Literature Review (SLR) in which Google Scholar was searched to identify key papers on the topic. From an initial 963 records (956 from string-based search and seven from a single backward snowballing procedure), only 15 relevant papers were identified.

### Results

Results show that current research is focused on the binary classification of Dengue, primarily using tree-based ML algorithms. Only one paper was identified using DL. Five papers presented solutions for multi-class problems, covering Dengue (and its variants) and Chikungunya. No papers were identified that investigated models to differentiate between Dengue, Chikungunya, and Zika.

**Funding:** VSS has a grant (062.00249/2020 [EDITAL N. 006/2019 - UNIVERSAL AMAZONAS]) from Fundação de Amparo à Pesquisa do Estado do Amazonas (FAPEAM) (http://www.fapeam.am.gov.br/). The funders had no role in study design, data collection and analysis, decision to publish, or preparation of the manuscript.

**Competing interests:** The authors have declared that no competing interests exist.

**Abbreviations: AI**, Artificial Intelligence; **ALP**, Alkaline phosphatase; **ALT**, Alanine transaminase; **ALYMPHP**, Atypical lymphocyte percent; **AST**, Aspartate aminotransferase; **AUC**, Area Under the Curve; **BIA**, Bioelectrical Impedance Analysis; **BRBES**, Specialised Belief Rules System; **CART**, Classification and Regression Tree; **CNN**, Convolutional Neural Network; **CZS**, Congenital Zika Syndrome; **DALYs**, disability-adjusted life years; **DBNs**, Deep Belief Networks; **DDM**, Dengue Diagnostic Model; **DENV**, Dengue Virus; **DF**, Dengue Fever; **DHF**, Dengue Hemorrhagic Fever; **DHF1**, Dengue Hemorrhagic Fever 1; **DHF2**, Dengue Hemorrhagic Fever 2; **DHF3**, Dengue Hemorrhagic Fever 3; **DL**, Deep Learning; **DSNs**, Deep Stacking Networks; **DSPM**, Dengue Severity Prediction Model; **DSS**, Dengue Shock Syndrome; **ELISA**, Enzyme-linked immunoassay; **EOSBAS**, Eosinophile basofile count; **EOSBASP**, Eosinophile basofile percent; **FLBES**, Fuzzy Logic Based Expert System; **FN**, False Negative; **FP**, False Positive; **FPR**, False Positive Rate; **GRU**, Gated Recurrent Units; **HAI**, Hemagglutination-inhibition assay; **HGB**, Hemoglobin; **HT**, Hypertension; **HTC**, Hematocrit; **ICU**, Intensive Care Unit; **ID3**, Iterative Dichotomiser 3; **IgG**, Immunoglobulin G; **IgM**, Immunoglobulin M; **JEV**, Japanese encephalitis virus; **kNN**, K-Nearest Neighbours; **LOO**, leave-one-out; **LSTM**, Long Short-Term Memory; **LYMPH**, Lymphocyte count; **LYMPHP**, Lymphocyte percent; **MCH**, Mean corpuscular hemoglobin; **MCHC**, Mean corpuscular hemoglobin concentration; **ML**, Machine Learning; **MLP**, Multilayer Perceptron; **MONO**, Monocyte count; **MONOP**, Monocyte percent; **MPV**, Mean platelet volume; **NCDs**, Non-communicable diseases; **NEUT**, Neutrophil count; **NEUTP**, Neutrophil percent; **NN**, Neural Networks; **NTDs**, Neglected tropical diseases; **OFI**, Other febrile illness; **PCR**, Polymerase chain reaction; **PDW**, Platelet distribution width; **PLCR**, Platelet large cell ratio; **PLT**, Platelet count; **RBC**, Red blood cells count; **RDW**, Red cell distribution width; **RNNs**, Recurrent Neural Networks; **ROC**, Receiver Operating Characteristic; **RT-PCR**, Reverse transcription polymerase chain reaction; **SLR**, Systematic Literature Review; **SOM**, Self Organising Map;

## Conclusions

The use of an efficient clinical decision support system for arboviral diseases can improve the quality of the entire clinical process, thus increasing the accuracy of the diagnosis and the associated treatment. It should help physicians in their decision-making process and, consequently, improve the use of resources and the patient's quality of life.

## Author summary

Neglected tropical diseases (NTDs) primarily affect the poorest populations, often living in remote, rural areas, urban slums or conflict zones. Arboviruses are a significant NTD category spread by mosquitoes. Dengue, Chikungunya, and Zika are three arboviruses that affect a large proportion of the population in Latin and South America. The clinical diagnosis of these arboviral diseases is a difficult task due to the concurrent circulation of several arboviruses which present similar symptoms and, sometimes, inaccurate test results. In this paper, we present the state of the art of studies investigating the automatic classification of arboviral diseases based on Machine Learning (ML) and Deep Learning (DL) models. Results show that current research is focused on the classification of Dengue, primarily using tree-based ML algorithms. The use of an efficient clinical decision support system for arboviral diseases can improve the quality of the entire clinical process, thus increasing the accuracy of the diagnosis and the associated treatment. It should help physicians in their decision-making process and, consequently, improve the use of resources and the patient's quality of life.

## 1 Introduction

Neglected tropical diseases (NTDs) include a wide range of parasitic, viral, and bacterial diseases that prevail in tropical and subtropical conditions in 149 countries and affect one billion people every year [1]. One major category of NTDs are arthropod-borne viruses (or arbovirus diseases), a group of viruses that are found in nature and biologically transmitted between susceptible vertebrate hosts by hematophagous arthropods [2].

Arboviruses included a wide variety of diseases including African swine fever virus, Japanese encephalitis virus (JEV), Rift Valley fever virus, tick-borne encephalitis virus, West Nile virus and yellow fever virus however the most common are Dengue, Chikungunya and Zika [3]. These three arboviruses are primarily transmitted by Aedes spp. mosquitoes, of which *Aedes aegypti* and *Aedes albopictus* are the most common vectors [4, 5]. The *Aedes aegypti* can easily adapt to urban and semi-urban areas [6, 7]. Population growth, unplanned urbanization, habitat modification, human and animal migration, and climate change, combined with low-quality housing and neglected peri-domestic environments, all contribute to creating ideal ecological conditions for urban Aedes spp. populations to thrive [5, 8–10]. These factors, disproportionately affecting the poor, increase the geographical area at risk to arboviral diseases and contribute to establishing arboviruses as a global health problem [8–11]. These arboviruses are maintained outside of rainy seasons by transovarial transmission from female mosquitoes to offspring [12, 13]. Other modes of transmission include vertical and sexual transmission [14, 15], and contaminated transfusions [5, 16]. The overall burden of arboviral diseases in general is significant. The incidence and number of deaths due to Dengue are increasing

**SVM**, Support Vector Machine; **TN**, True Negative; **TORCHS**, Toxoplasmosis, Rubella, Cytomegalovirus, Herpes Symplex, and Syphilis infections; **TP**, True Positive; **TPR**, True Positive Rate; **WBC**, White blood cells; **XAI**, Explainable Artificial Intelligence; **ZIKV**, Zika Virus.

resulting in a global burden of disease of 2.9 million disability-adjusted life years (DALYs) for 2017 alone, a 107% increase since 1990 [17]. Recent analysis on the global burden of Chikungunya and Zika suggest an average yearly loss of over 106,000 and 44,000 DALYs, respectively, between 2010 and 2019 [18]. In each case, the burden of these diseases disproportionately impacts the Americas.

While the clinical presentation of these diseases are well-established [19, 20], diagnosing these diseases is a difficult task. Three primary reasons are cited in the literature to explain why there are difficulties in making an arboviral diagnosis. Firstly, the majority of cases are asymptomatic thus arbovirus may be present in an area without an identifiable outbreak [21, 22]. Secondly, their symptomatic infection is usually clinically indistinguishable from each other. All of them share common symptoms like fever, arthralgia, myalgia, headache, and retro-orbital pain [21]. While Dengue and Zika have some distinct symptoms, for example, hemorrhagic diathesis (Dengue) and edema in limbs (Zika), and Chikungunya is related to joint complaint, their diagnosis requires a high degree of experience and clinical insight which can be further complicated in special populations [22, 23]. In addition, Dengue and Chikungunya symptoms may include hemorrages and leukopenia/thrombocytopenia, while Chikungunya and Zika symptoms may include non-purulent conjunctivitis [22]. Thirdly, co-infection is also common thus increasing the difficulty of diagnosis of these conditions [5, 21, 24, 25].

Despite the difficulties in differential diagnosis, the progression and impact of these diseases varies significantly. After infection by Dengue, the disease may manifest asymptomatically and patients may not even know that they are infected. Serologically, after 7-10 days after the mosquito bite, a diagnosis of Dengue can be confirmed [26]; some people may experience symptoms such as fever, headache, pain in the muscles and joints, and fatigue. For some, the disease may progress to a more severe condition resulting in bleeding, organ damage, and plasma leakage [19]. Dengue can be classified into two stages—the febrile phase and the critical phase. The febrile phase typically lasts for 2-7 days. The critical phase of dengue begins at defervescence and typically lasts 24–48 hours. While most patients clinically improve, some may experience systemic vascular leakage syndrome, characterised by increasing hemoconcentration, hypoproteinemia, pleural effusion, and ascites [27]. Severe Dengue can result in death due to plasma leaking, fluid accumulation, respiratory distress, severe bleeding, or organ impairment [19]. Chikungunya infection may manifest symptoms similar to Dengue between the fourth and seventh day after the bite, but with greater joint pain. The progression of Chikungunya has three phases. The acute phase is characterised with sudden onset symptoms manifesting with high fever, rash, and arthralgia, affecting mainly the small and large joints. The subacute phase is characterised by worsening arthralgia. While Chikungunya is rarely fatal, it can progress to a chronic state. Post-Chikungunya rheumatism is common and can last from weeks to years with associated adverse effects on quality of life [5, 28–31]. Zika was initially considered a mild disease sometimes with no fever episode [5], however it is now clear that its major threat is related to microcephaly and other congenital abnormalities in the fetus and newborn; it may trigger Guillain-Barré syndrome, neuropathy, and myelitis in adults and older children [31]. Symptoms of Zika include arthralgia, edema of the extremities, low fever, maculopapular rash that is often pruritic, headaches, retro-orbital pain, without purulent conjunctivitis, vertigo, myalgia, and digestive disorders [32]. The most serious manifestation of infection is Congenital Zika Syndrome (CZS). The risk of the infection can occur during any gestational trimester [33]. CZS is related to fetal microcephaly, fetal brain disruption sequence, subcortical calcifications, pyramidal and extrapyramidal signs, ocular abnormalities (focal pigmented mottling, chorioretinal atrophy), congenital contractures, fetal growth restriction, and even death [33–35].

Early identification of specific arbovirus infections can have a significant impact on the clinical course and decisions related to treatment and care. The adverse impacts of poor

arbovirus diagnosis are exacerbated where there are competing pressures for funding and trained and experienced staff, due to multiple concurrent disease epidemics [36]. Novel low-cost scalable approaches to the differential diagnosis of arboviral diseases for epidemiological surveillance are required. One such approach is the development of computational models for monitoring and diagnostic classification based on clinical data and symptoms. Machine Learning (ML) and Deep Learning (DL) models have been widely proposed in the biomedical field to support the diagnosis and prediction of disease [37]. ML is a computational method that makes use of experience to make predictions, i.e., it is an algorithm that receives input data (training data set) to learn or find a pattern. Data quality and size are fundamental to the success of the learning process and as consequence to guarantee the efficiency of the model predictions. When designing a ML model, the goal is to find a configuration (a set of hyper-parameters) that produces a model able to generalise and produce satisfactory performance when dealing with previously unseen new data. DL is a sub-field of ML which emphasises learning based on successive layers of increasingly meaningful representations [38]. Here, "deep" is related to the idea of successive layers of representations. DL models are based on early iterations of Neural Networks (NN) and are increasingly reported as the most effective ML approach with the advantage of combining the feature extraction and the classification task at the same time. However, the "black box" nature of most DL models is a significant challenge in the health space which values transparency. As such more transparent ML models are commonly used due to their interpretability.

In this paper, we present an SLR on how existing research employs ML and DL techniques to automatically classify arboviral diseases and support clinical diagnosis.

## 2 Methods

The purpose of an SLR is to identify, select and critically appraise research on a specific topic. SLRs typically comprise three main phases: planning the review, conducting the review, and reporting the review results [39]. The goal of this paper is to present evidence on the state of the art of studies investigating the automatic classification of arboviral diseases to support clinical diagnosis based on ML and DL models. To accomplish this goal, this SLR follows the methodology present in Fig 1 and seeks to address the following research questions:

- **RQ 01**: What arboviruses are the focus of research on ML and DL classification of arboviral diseases to support clinical diagnosis?

- **RQ 02**: Which ML and DL techniques are being used in research relating to the classification of arboviral diseases to support clinical diagnosis?

- **RQ 03**: How are ML and DL models being designed and how do they perform when classifying arboviral diseases?

- **RQ 04**: What data characteristics are considered when applying the ML and DL techniques?

- **RQ 05**: What are the metrics being used to evaluate the performance of the ML and DL techniques?

### 2.1 Search strategy

The search strategy comprised an automated and manual phase. A literature search was conducted using Google Scholar with the following search string: *(("deep learning" OR "machine learning") AND ("arbovirus" OR "arboviral") AND ("classification" OR "diagnosis" OR "analysis") AND ("clinical data"))*, in March, 2021. Google Scholar was selected due to the

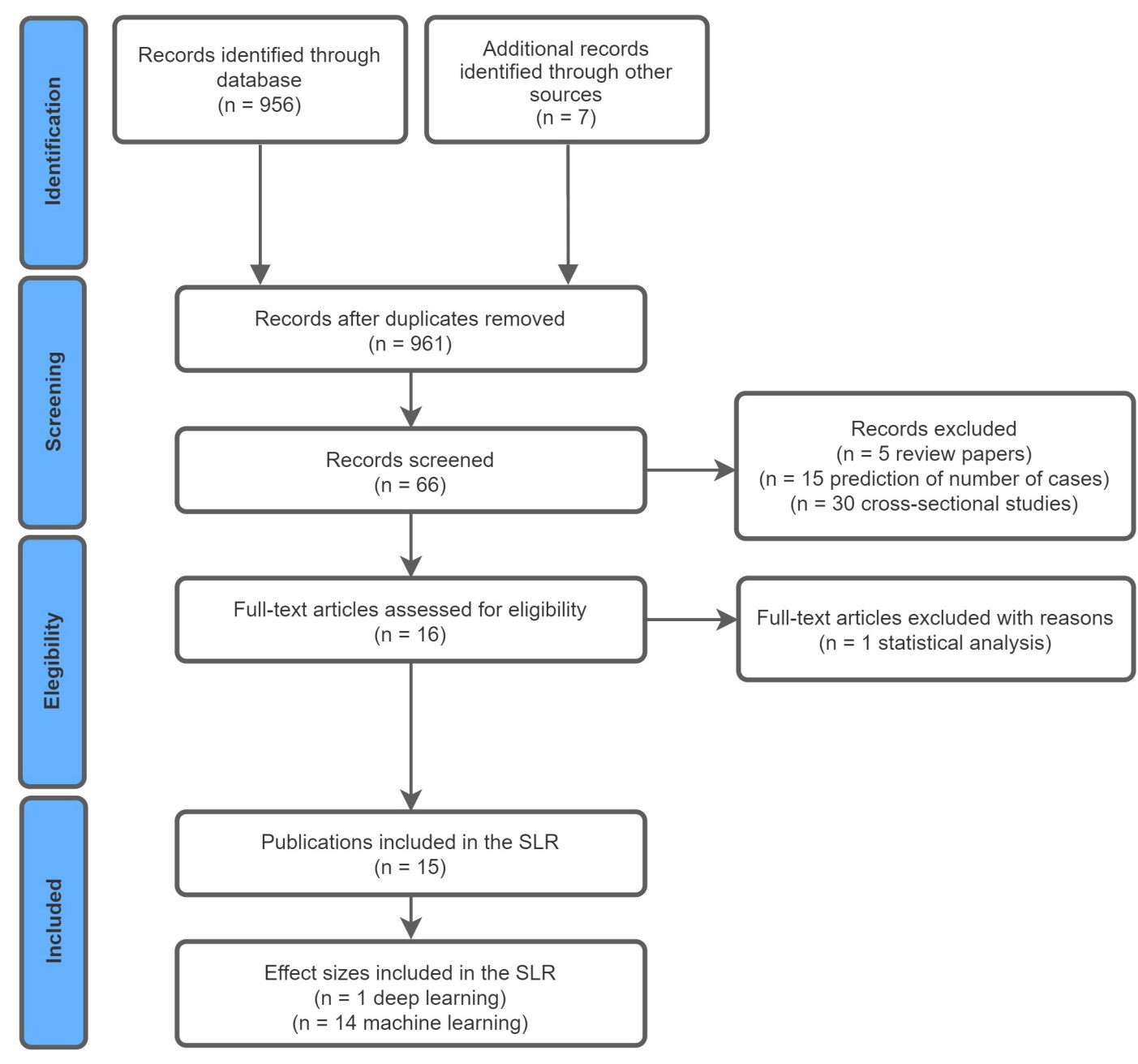

**Fig 1. Study selection.**

comprehensiveness of articles indexed in its database [40]. The authors performed quality checks on other common academic and bibliometric databases, to ensure no major articles were left out.

Wohlin [41] defines snowballing as "...*the usage of a reference list of papers or citations to the paper to identify additional papers*". There are two ways to use this procedure: backward and forward. The backward snowballing uses the reference list to find new papers to include in the systematic review, while the forward snowballing identifies new papers based

on those papers that cite the relevant papers. To expand the sample of papers in the SLR, we performed a manual search to identify and download relevant studies from a single iteration reverse snowballing procedure applied to relevant studies found in the automated search.

## 2.2 Study selection

To ensure the selection of only studies relevant to our review, we consider studies that meet specific inclusion criteria. The inclusion criteria required that papers be in the English language and use clinical data in the application of ML or DL models for arbovirus diagnosis in a primary study. Papers were excluded if they *(1)* did not examine the diagnosis of arboviruses, *(2)* were in a language other than English, *(3)* were a secondary or tertiary study, *(4)* used conventional statistical techniques, or *(5)* did not use clinical data as inputs to the ML and DL models.

The initial automated search returned 956 records. These had their title and abstracts assessed by two independent authors according to the inclusion and exclusion criteria. Where a conflict arose, a third author arbitrated on selection. Based on the inclusion and exclusion criteria, nine papers were retained in the sample. We performed a single backward snowballing procedure. This returned a further seven papers, six of which were retained in the sample following review. The final sample was 15 papers.

## 2.3 Data extraction and coding

The following data was extracted for each study: authors, publication year, arboviral disease type(s), ML and DL technique(s) employed, the data set used in the study, the data characteristics used as input, and metrics used to evaluate the ML and DL performance.

## 3 Results and discussions

### 3.1 What arboviruses are the focus of research on Machine Learning and Deep Learning classification of arboviral diseases to support clinical diagnosis?

Surprisingly, given the range of arboviral diseases, the focus of research, albeit a small sample, was the three most popular diseases i.e. Dengue, Chikungunya, and Zika. Although these three are common arboviral diseases, no studies were found that carried out multi-classification considering between these three arboviral diseases or other arboviral types, such as West Nile virus, yellow fever virus, Saint Louis Encephalitis virus, Mayaro virus, Oropouche virus and others, showing that there is space for further investigations in this area.

Most of works presented models for binary classification—Dengue or not [42–47]; Dengue Hemorrhagic Fever (DHF) or not [48]; Chikungunya or not [49]; and Zika classified between "Discarded cases" and "Somewhat probable" for CZS [50]. It is interesting to note that the only work that deals with Zika is focused on CZS, not covering Zika in general.

Four studies sought to address multi-class problems. Thitiprayoonwongse et al. [51] classified between Dengue Fever (DF), Dengue Hemorrhagic Fever 1 (DHF1), Dengue Hemorrhagic Fever 2 (DHF2) and Dengue Hemorrhagic Fever 3 (DHF3). Fahmi et al. [52] focused only on Dengue, classifying between DF, DHF and Dengue Shock Syndrome (DSS). Veiga et al [50] classified between "Discarded cases", "Somewhat probable", "Moderately probable" and "Highly probable" of having CZS, while Lee et al. [53] proposed models to differentiate between DF, DHF and Chikungunya.

## 3.2 Which Machine Learning and Deep Learning techniques are being used in research relating to the classification of arboviral diseases to support clinical diagnosis?

ML has basically four categories of learning techniques: supervised learning, unsupervised learning, semi-supervised learning, and reinforcement learning. The techniques identified in this SLR are based on supervised learning. Supervised learning systems solve the function approximation problem in which the training data is a set of (x, y) pairs and the aim is to produce a prediction $y^*$ in response to a $x^*$ [54]. Supervised classification is one of the most frequent tasks performed and unsurprisingly a large number of techniques leverage ML including Decision Trees, NN, Support Vector Machine (SVM), Naive Bayes, Logistic Regression and K-Nearest Neighbours (kNN)) [55].

Despite the advantages and wide applicability of ML models, they suffer from selectivity-invariance issues [56]. Such issues limit their capacity to process raw data thus requiring careful (and time consuming) feature selection engineering before model training. DL (also known as deep structured learning, hierarchical learning, or deep ML) models negate this ML issue. DL models are composed of multiple abstraction layers that are able to automatically learn from features of the original data thereby removing the need for feature selection. The usage of non-linear models facilitate the discovery of solutions for more complex problems. Commonly, DL architectures have many hidden layers composed of neurons connected through an activation function. Sub-section 3.2.7 describes a common DL, the Convolutional Neural Network (CNN).

Fig 2 presents the ML and DL techniques used to perform arboviral diseases classification in the SLR sample (for the purpose of this paper, we consider traditional statistical techniques (e.g. Decision Trees, Logistic Regression and Naive Bayes) as ML as per [57]. Only one paper in the SLR sample used DL. Ho et al. [47] compared a CNN with Decision Tree and Logistic Regression models. All other papers employ common ML techniques, including tree-based algorithms (Decision Trees, Random Forest, AdaBoost and Gradient Boost), NN, SVM, Naive Bayes, Logistic Regression, and kNN.

**3.2.1 Tree based algorithms: Decision Tree, Random Forest, AdaBoost and Gradient Boost.**  A Decision Tree is a non-parametric method that can be applied in problems with categorical variables (classification tree, the focus of this work) and also with continuous variables

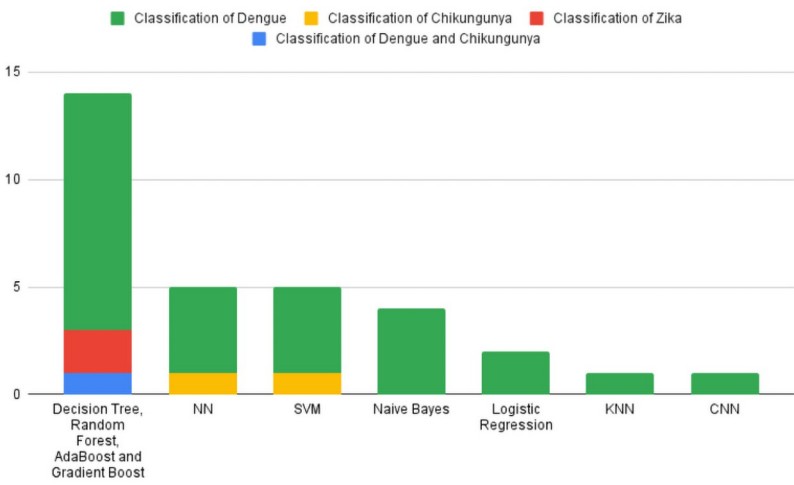

**Fig 2. Models used in the works divided by the main problems.**

(regression tree). A tree is composed of a root node, internal nodes, and leaf nodes, and it is built successively dividing data according to one of the predictor variables [58]. To build a Decision Tree, it is necessary to define the node-splitting algorithm to minimise the impurity of the node. If the split achieves the maximum reduction of impurity, then the node is defined as a leaf [59]. The most common splitting algorithms are the Information Gain (used by Classification and Regression Tree (CART)) and the Gini index (used by Iterative Dichotomiser 3 (ID3) and C4.5 algorithms). The main advantage of using Decision Tree algorithms is the implicit feature selection during the model building process and the interpretability of results. Decision Trees are also able to handle missing values, which are commonly encountered in clinical studies [42]. On the other hand, over-complex trees do not generalise the data well, often presenting overfitting (or underfitting), and are prone to errors with a relatively small number of samples for training. The SLR sample includes eleven Decision Tree models—[42, 44–47, 50–53, 60, 61].

In contrast to Decision Trees (sometimes referred to as 'strong learner' approaches) optimised to solve a specific problem by looking for the best possible solution, ensemble learning techniques are based on a set of "weak learners". Ensemble learning techniques can be categorised into three classes: *(1)* bagging (or bootstrapping), *(2)* boosting, and *(3)* stacking [62]. Random Forest is an ensemble technique based on bagging that combines several Decision Trees. It is built randomly from a set of possible trees with $K$ characteristics in each node. Random in this context means that in the set of trees, each tree has an equal chance of being sampled. Multiple classification trees are obtained from bootstrap samples in order to calculate the final majority classification. The SLR sample includes three Random Forest models—[48, 50, 52]. As Random Forest models combine different Decision Trees, their results are not as easy to understand as a Decision Tree and are also more expensive computationally. Notwithstanding this, Random Forests typically outperform Decision Trees and handle balancing errors better when working with an imbalanced data set [63].

Boosting is an ensemble technique that combines $k$ low performance models ($M1$, $M2$..., $Mk$) in order to improve the final model, $M^*$ [64]. The $k$ classifiers are learned iteratively and after a $Mi$ is learned, the weights are updated in order to generate the next classifier, $Mi + 1$. Performance is improved by training tuples that were misclassified by $Mi$. The final boosted model, $M^*$, combines the results of each $k$ classifier. The Adaptive Boosting (AdaBoost) [65] is the first stepping stone in boosting techniques and it uses Decision Trees with a single split (one node and two leaves), also named Decision Stumps, as "weak learners". Gradient Boost uses a technique named forward stage-wise additive modelling that adds a new Decision Tree at each step to minimise a global cost function using the Steepest Gradient Descent method [62]. The main advantages of boosting algorithms in general, including AdaBoost and Gradient Boost, are intrinsic automated variable selection, and flexibility regarding the type of predictors and stability when handling high-dimensional data [66]. AdaBoost, in particular, is also known to be quite resistant to overfitting. While these advantages have attracted the attention of biomedical researchers [66], only one paper in the SLR sample proposed an AdaBoost model (Fahmi et al. [52]) and another, Veiga et al. [50], proposed a Gradient Boost model.

**3.2.2 Support Vector Machine (SVM).** SVM is a classifier based on Vapnik's statistical learning theory [67]. To perform classification, SVM builds hyperplanes in a multidimensional space in order to separate instances of different classes. The goal is to find the optimal separating hyperplane and, at the same time, maximise the distance between the support vectors (which are the extreme delimiters) [62, 67].

Robustness is one of the main advantages of SVM models. Data with outliers do not impact negatively in SVM model performance. While Decision Tree models benefit from interpretability, lack of transparency is a drawback of SVM models, especially when dealing with

high-dimensional data sets. SVM models can also be quite memory-intensive and therefore processing large and complex data sets can be slow [62]. SVM models feature in five studies in the SLR sample—[43, 46, 48, 49, 52].

**3.2.3 Neural Networks (NN).**   Inspired by the human nerve system, an NN is composed of groups of nodes (or units) simulating layers of neurons. Each neuron is multiplied by weights to simulate synapses and the result is passed to the neuron of the next layer. All results received by a neuron are summed up and a mathematical formula, called an activation function, is used to convert the received value to be transported to the next layer of neurons, simulating the activation of a human neuron. This type of architecture allows the training of a NN model by adjusting the weights that connect neurons through a learning experience. During the NN training, different sets of neurons in the NN are "activated" and at the end, the goal is that the model can generalise past patterns.

The most basic NN model is a perceptron [68]. It is composed of only two layers of neurons: *(1)* the input layer to receive the data, and *(2)* the output layer to perform the prediction. Once it has a simple activation function, it is possible to solve linearly separable problems. The Multilayer Perceptron (MLP) is an evolved form of perceptron with additional layers of neurons in the middle, the hidden layers, and a more complex activation function in the neurons. These additions make the MLP very useful to solve complex problems for both classification and regression. MLPs are often referred to as NN and the terms are used interchangeably. In general, NNs present many advantages including high capacity to learn and generalise, and the ability to deal with imprecise, fuzzy, noisy, and probabilistic information [69, 70]. As such, they are widely used in health research [71–73].

MLP was a popular ML solution in the 1980s with applications in various fields. Recently the interest in this type of model was renewed due to the success of DL. Several authors classify MLP as a traditional model of ML [74, 75], but with the advent of DL, concepts of MLP were improved and it can also be classified as a DL [76]. In this paper, we classify MLP as a traditional model of ML due to the context observed in the selected proposals. Five papers employed NN models in the SLR sample: [44, 45, 49, 52, 77].

**3.2.4 Naive Bayes.**   Naive Bayes is a probabilistic classifier that performs classification based on the Bayes' Theorem, selecting the most likely class according to its independent variables [78]. The term *naive* is due to the way the model calculates the probabilities of each event i.e. all attributes of the data set are equally important and independent.

In general, Naive Bayes models are simple, fast and effective, and perform well when a data set contains outliers or is missing data [62], common features in many health data sets. However, Naive Bayes models are not without drawbacks. They assume that all attributes of a data set have the same importance which is often not true. If a data set has large numbers of attributes, the reliability of the results may be limited. Four works in the SLR sample applied Naive Bayes models—[43, 45, 48, 52].

**3.2.5 Logistic regression.**   Logistic Regression is a classification technique based on idea of modelling the odds of belonging to Class 1 using an exponential function [62]. In this technique, the dependent variable, $Y$, is binary and the independent variables, $X = \{x_1, x_2, \ldots, x_n\}$, are used to estimate the value of $Y$ by using a logistic function. The goal is to find an optimal hyperplane, that separates the two target classes (binary classification). In case of multi-class problems, the one-vs-all strategy can be used to address the problem.

Some advantages of this model include dealing with categorical independent variables and a high degree of reliability. As disadvantages, this type of model does not generalise well when using a large number of features, it is vulnerable to overfitting, and cannot solve non-linear problems, requiring a transformation of non-linear resources [79]. Two works employed Logistic Regression in the SLR sample—[47] and [52].

**3.2.6 k-Nearest Neighbours (kNN).**   kNN is a classification technique that defines a minimum number of neighbours, *k*, and calculates the distance, the similarity, of each data element with respect to its *k* neighbours. There are many measures of similarity, and the most commons approaches are the Euclidean distance and the Minkowski metric [62]. The most frequent class among the neighbouring *k* is determined as the class of the target instance [80].

Simplicity is the main advantage of kNN. kNN is a good choice when working with a small low-dimensional data set however it can be extremely inefficient when dealing with large data sets because it is computes all pairwise distances [62]. Only one paper in the SLR sample proposed a kNN model, Fahmi et al. [52], using Euclidean distance with $k = 5$.

**3.2.7 Convolutional Neural Networks (CNN).**   A CNN is a DL technique. CNNs are designed to process input data in the form of multiple arrays [81]. A basic CNN comprises convolutional layers, pooling layers, nonlinear function (generally ReLU), and fully connected layers. Units in a convolutional layer are organised into feature maps and each unit is connected to local patches in the feature maps of the previous layer. It is done by using a set of weights called filters. The result of this local weighted sum is passed through a nonlinear activation function. All units in a feature map share the same filter bank, and different feature maps in one layer can use different filter banks. The result of this whole process feeds fully connected layers resulting in a final classification. As discussed earlier, DL models, including CNN, often outperform traditional ML models however their adoption in health settings has suffered due to an inherent lack of transparency.

Although the methodology of training and testing models is clearly well-defined, the resultant models themselves can be often unexplainable to humans [82]. Even when techniques are used to select attributes resulting in good model performance, the relationships between those attributes and the output classification may not directly track causal relationships in the real world [82]. One paper in the SLR sample, Ho et al. [47], uses a CNN, DenseNet. DenseNet is a CNN architecture in which each layer is connected to all others within a dense block [83]. In this case, all layers can access feature maps from their preceding layers enabling heavy feature reuse. As a direct consequence, the model is more compact and less prone to overfitting. Furthermore, each individual layer receives direct supervision from the loss function through the shortcut paths, which provides implicit deep supervision [84].

## 3.3 How are Machine Learning and Deep Learning models being designed and how do they perform when classifying arboviral diseases?

As presented in Section 3.1, studies included in the SLR sample focused on only three arboviruses—Dengue, Chikungunya, and Zika. Before detailing the models, we present some basic concepts to better understanding the proposals in the sample.

Most of ML and DL models have a set of hyperparameters that can be adjusted to achieve a better performance [85]. There are basically two types of hyperparameter optimisation methods: manual search and automatic search [86]. In manual research, testing is done manually based on the basic intuition and experience of users in identifying the important parameters, i.e. those that have the greatest impact on the results. The best known automatic search method is the grid search. In this method, the combination of all possible values of hyperparameters is defined given a range; their performance is compared according to some predefined metric, and the configuration that achieves the best performance is selected. Although this method is widely used and presents interesting results, it suffers from dimensionality, i.e., the efficiency of the algorithm rapidly decreases as the number of hyperparameters being adjusted and the range of hyperparameter values increase.

Given this disadvantage of grid search, the random search algorithm is an alternative which reduces the search for hyperparameters, pointing to an approximate solution of an optimisation function, i.e. random combinations are performed within a range of values. By comparing random search with grid search, we have increased efficiency in processing time however as the search space is limited, better solutions may be excluded [86].

Another process widely used in ML is the selection of attributes for the construction of a model, so-called feature selection. Feature selection can be divided into three categories: wrapper, filter, and embedded [87]. Wrapper methods use the predictive performance of a predefined learning algorithm to assess the quality of selected attributes. With the selection of a specific learning algorithm, the typical wrapper method performs two steps: *(1)* it looks for a subset of attributes, and *(2)* it evaluates the selected attributes. It repeats the steps until some stopping criteria is met. Filter methods are independent of the learning algorithm. They look at data characteristics to assess the importance of a given attribute. This method is generally more computationally efficient but due to the lack of a specific learning algorithm in the attribute selection step, the selected features may not be ideal for the target learning algorithm. Finally, embedded methods are a trade-off between filter and wrapper methods, incorporating feature selection into model learning. This method inherits the advantages of wrapper and filter methods including interactions with the learning algorithm. As such, it is much more efficient than wrapper methods as it does not need to iteratively evaluate sets of attributes. Basically, this method proposes to reduce the computation time used to reclassify different subsets by incorporating the selection of resources as part of the training process [88].

In ML and DL modelling, overfitting is a common problem. According to Ying [89], overfitting happens when the model does not generalise the observed data well with the unseen data i.e. the model fits very well in the training set but fits poorly in the test set. Overfitting can occur because noise may be present in the training set. This can happen when the training set is very small or when data is not very representative or there is too much noise. Overfitting may also occur when the algorithm has too many hypotheses (many inputs) thus compromising the balance between the accuracy and consistency of the learned model. In this case, the model does not handle different source data sets well.

There are at least four approaches to addressing the overfitting problem: early-stopping, network-reduction, expansion of the training data, and regularization [89]. Early stopping involves stopping training at the point that performance on a test data set starts to degrade. This phenomenon is known as "learning speed slow-down". In this case, the model continues to learn after a given point, increasing the validation error and consequently decreasing the training error. Two problems can arise. Firstly, stopping before the best result point thus making the result sup-optimal. Secondly, stopping after the point with the best result which results in an over-adjustment. Therefore, the main objective of this solution is to find the exact point that training should be interrupted [89, 90].

In network-reduction, the concept of pruning is used, reducing the classification complexity by eliminating less significant or irrelevant data, preventing overfitting and improving classification accuracy. There are two standard pruning approaches: pre-pruning and post-pruning. In pre-pruning, its functioning takes place during the learning process, using, for example, stopping criteria through some rule (length, cost, significance test etc.). In post-pruning, the training set is divided into two sets: crescent set and pruning set. This approach ignores the overfitting problems during the learning process in the growing set, so they avoid overfitting by excluding rules from the model generated during learning [89, 91].

Adjusting hyperparameters brings balance and regularity in model training, but to make these adjustments sufficient samples are needed for learning. In this sense, an expanded data set can greatly improve the accuracy of these models. This type of approach has been widely

used to improve the generalisation performance of models. Commonly, four approaches are used to expand a data set: *(1)* acquire more training data; *(2)* add some random noise to an existing data set; *(3)* re-acquire some data from an existing data set through some form of additional processing; and/or *(4)* produce new data based on the distribution of the existing data set [86, 89]. Expanding data sets has drawbacks. With greater data, there will consequently be an increase in training time. In addition, such data can be difficult to acquire and often require human intervention for labelling data.

Finally, regularisation is a technique that applies a small variation to the original data to efficiently train a model such that the model generalises better [92]. To carry out this process another term is added, a penalty, also known as regularizer. There are three main regularisation methods: L1 regularisation, L2 regularisation, and dropout. With L1 regularisation, the absolute value of the weights is penalised. L1 regularisation is also known as the least absolute shrinkage and selection operator (LASSO) [92]. In contrast, L2 regularisation defines the weights of each feature and removes features from the model, only keeping the most valuable features. This allows the model to be simpler and more interpretable. This regularisation method uses the Euclidean distance as a penalty term. The dropout method is a solution to avoid overfitting in neural networks. Basically, it randomly eliminates units and connections in the network during the training process. This process usually takes place using the following steps: *(1)* releasing half of the randomly hidden neurons and building a simpler network; *(2)* training the simple network using a stochastic descending gradient; *(3)* restoring the neurons removed in Step 1; *(4)* removing half of the hidden neurons from the new network to form a new simple network; and, *(5)* repeating the entire process until the ideal set of parameters is reached [89, 93, 94].

Table 1 presents a list of included studies by year of publication, classification target, ML and/or DL techniques used, model configurations, software, evaluation metrics, and optimisation techniques employed for hyperparameter and feature selection.

**3.3.1 Dengue.**    Of the 15 relevant papers in the SLR sample, 12 included the diagnosis of Dengue in their studies including binary classification and multi-class classification:

- **Binary classification**

  - **Dengue or not Dengue**: Tanner et al. [42], Fathima and Hundewale [43], Sajana et al. [44], Gambhir et al. [45], Sanjudevi and Savitha [46], Ho et al. [47];

  - **Severity of Dengue**: Tanner et al. [42], Potts et al. [60], Phakhounthong et al. [61];

  - **Risk of Dengue**: Faisal et al. [77]; and

  - **DHF or not**: Arafiyah et al. [48]
    Tanner et al. appeared twice because they presented two distinct classification problems.

- **Multi-class classification**

  - **DF, DHF or DSS**: Fahmi et al. [52]; and

  - **DHF1, DHF2 or DHF3**: Thitiprayoonwongse et al. [51]

Tanner et al. [42] proposed two Decision Tree models. First, the Dengue Diagnostic Model (DDM) sought to classify if a patient had Dengue or not using 1,200 records of patients with acute febrile illness. The second, the Dengue Severity Prediction Model (DSPM), sought to classify the severity of Dengue in adults using data from 161 patients. A C4.5 Decision Tree classifier was built using the Inforsense software. A k-fold cross-validation approach ($k = 10$) was used to avoid model over-fitting. Both models presented good performance however the

**Table 1. Overview of the primary studies.** 干: The study seems to present a model overfitting or an inappropriate benchmarking methodology.

| Primary studies | Year | Target classification | ML and/or DL | Model configuration | Software | Metrics | Hyperparameter optimisation | Feature selection |
|---|---|---|---|---|---|---|---|---|
| Tanner et al. [42] | 2008 | (1) Dengue or not (2) Severity of Dengue | Decision Tree | Not described | Not described | Sensitivity, specificity error rate, AUC | Not applied | Decision Tree |
| Fathima and Hundewale [43] | 2012 | Dengue or not | Naive Bayes | Default values from the package e1071 | R | Accuracy, sensitivity, specificity, rate risk | Not applied | Not applied |
| | | | SVM | Not described | | | Grid Search | |
| Sajana et al. [44] 干 | 2018 | Dengue or not | Decision Tree | Not described | Not described | Accuracy, precision, recall, F-Measure | Not applied | Not applied |
| | | | NN | Not described | | | | |
| Gambhir et al. [45] | 2018 | Dengue or not | NN | hidden layers = 3 lr = 0.3 momentum = 0.25 | Not described | Accuracy, sensitivity specificity, error rate | Not applied | Not applied |
| | | | Decision Tree | criterion = Information Gain size of split = 2 min leaf size = 2 min gain = 0.01 max depth = 20 confidence = 0.5 | | | | |
| | | | Naive Bayes | estimation method = greedy min bandwidth = 0.01 num of kernels = 10 | | | | |
| Sanjudevi and Savitha [46] 干 | 2019 | Dengue or not | Decision Tree | Not described | WEKA | Accuracy, sensitivity, specificity, AUC | Not applied | Not applied |
| | | | SVM | Not described | | | | |
| Ho et al. [47] | 2020 | Dengue or not | Decision Tree | criterion = gini min leaf size = 20 xval = 10 cp = 0.01 | Not described | ROC, AUC | Not applied | Crude odds ratios Adjusted odds ratios |
| | | | Logistic Regression | solver = lbfgs | | | | |
| | | | CNN | hidden layers = 16 | | | | |
| Potts et al. [60] | 2010 | Severity of Dengue | Decision Tree | criterion = gini min samples split = 0.05 max depth = 5 min impurity decrease = 0.0001 | SPSS Answer Tree 3.0 | Sensitivity, specificity | Not applied | Decision Tree |
| Phakhounthong et al. [61] | 2018 | Severity of Dengue | Decision Tree | Not described | WEKA | Accuracy, sensitivity, specificity | Applied but not described | Logistic Regression |
| Faisal et al. [77] | 2010 | Risk of Dengue | NN | neurons = 10 lr = 0.1 momentum = 0.99 iterations = 20.000 | Not described | Accuracy | Grid Search | SOM |
| Thitiprayoonwongse et al. [51] | 2012 | DF, DHF1, DHF2 or DHF3 | Decision Tree | (1) confidence = 0.4 | Not described | Accuracy, sensitivity, specificity | Not applied | Decision Tree |
| | | | | (2) confidence = 0.3 | | | | |

*(Continued)*

**Table 1.** (Continued)

| Primary studies | Year | Target classification | ML and/or DL | Model configuration | Software | Metrics | Hyperparameter optimisation | Feature selection |
|---|---|---|---|---|---|---|---|---|
| | | | | (3) Not described | | | | |
| Arafiyah et al. [48] | 2018 | DHF or not | Random Forest | Not described | Orange | Accuracy, sensitivity | Not applied | Not applied |
| | | | SVM | Not described | | | | |
| | | | Naive Bayes | Not described | | | | |
| Fahmi et al. [52] | 2020 | DF, DHF or DSS | NN | neurons = 100 activation = Relu solver = Adam reg alfa = 0.0001 iterations = 200 | Orange | Accuracy, sensitivity, precision | Not applied | ReliefF |
| | | | Decision Tree | criterion = Information gain min leaf = 5 min instances = 2 max deph = 100 | | | | |
| | | | SVM | c = 100 kernel = rbf tolerance = 0.0010 max iteration = 100 | | | | |
| | | | KNN | k = 5 distance metric = euclidean weight = uniform | | | | |
| | | | Random Forest | n estimators = 10 size of split = 5 criterion = gini | | | | |
| | | | Naive Bayes | Not described | | | | |
| | | | AdaBoost | base estimator = Decision Tree n estimators = 50 lr = 1.0 algorithm = SAMME.R loss function = linear regression | | | | |
| | | | Logistic Regression | Default configuration | | | | |
| Hossain et al. [49] | 2019 | Chikungunya or not | NN | Not described | Matlab | AUC | Not applied | Not applied |
| | | | SVM | Not described | | | | |
| Veiga et al. [50] | 2021 | Zika (CZS) or not | Random Forest | n estimators = 100 max depth = 5 size of split = 40 | | Sensitivity, precision F1-score | Grid search | Applied but not described |
| | | | Gradient Boost | max depth = 8 size of split = 5 | | | | |
| | | | KNN | Not described | | | | |
| | | | Decision Tree | Not described | | | | |
| | | | Adaboost | Not described | | | | |
| Lee et al. [53] | 2012 | DF, DHF or Chikungunya | Decision Tree | Binary recursive partitioning | R | Sensitivity, specificity AUC | Pruning | Logistic Regression |

DDM performed better across all metrics, i.e., sensitivity (71.2%), specificity (90.1%), overall error rate (15.7%) and Area Under the Curve (AUC) (0.88). This is unsurprising given the larger data set available to the DDM.

Fathima and Hundewale [43] compared two classification models, SVM and Naive Bayes, to classify if a patient had Dengue or not. To determine the best SVM hyperparameters, a Grid Search was performed changing the gamma parameter and the cost (c). Despite executing the Grid Search, neither the best configuration nor the configuration of the Naive Bayes model was detailed. In general, the SVM model presented the best performance, despite its low sensitivity (47%). The Naive Bayes model presented high sensitivity and very low specificity, accuracy and risk rates, all above 18%.

Sajana et al. [44] proposed three models for binary classification of Dengue using clinical and laboratory data: an MLP, and two Decision Trees (C4.5 and CART). Like Fathima and Hundewale [43], the configurations of the models were not detailed. The CART model presented the best results, achieving 100% in all metrics (accuracy, sensitivity, precision and F-Measure). There is no mention regarding the use of feature selection and hyperparameter optimisation. Given the results, it is possible that the models were overfitting due to the small amount of data available, i.e., only 20 records.

Gambhir et al. [45] proposed three models (NN, Decision Tree and Naive Bayes) to classify whether a patient had Dengue or not. The configuration of the models were described in the paper and are summarised in Table 1. K-fold cross-validation ($k = 10$) was used to validate and test the models. The NN presented the best results—79.09% accuracy, 55.55% sensitivity, and 88.5% specificity. However, the other models achieved similar performance. Gambhir et al. [45] did not describe whether hyperparameter optimisation or feature selection was applied in the data set.

Sanjudevi and Savitha [46] compared Decision Tree and SVM models to classify whether a patient had Dengue or not. Model configurations were not described. The WEKA tool was employed to run the experiments and calculate the metrics. The SVM model obtained the best results achieving 100% sensitivity, 100% specificity, 100% precision, and AUC of 99%. The extremely high performance results suggest model overfitting.

Ho et al. [47] proposed three models to classify Dengue using clinical data—a Decision Tree, a Logistic Regression, and a CNN. Models were validated and tested using k-fold cross-validation ($k = 10$). Feature selection was performed using crude odds ratio and adjusted odds ratio analysis. From 18 available attributes, four were initially selected—age, body temperature, White blood cells (WBC) count, and Platelet count (PLT). Three experiments were performed with more attributes: *(1)* six attributes, the four previously mentioned plus gender and haemoglobin count; *(2)* 11 attributes, the previous six and five more vital signals; and *(3)* the entire data set with 18 attributes. Results suggested that when using only four attributes, the AUC in all experiments were close to 84%. The CNN performed marginally better than Decision Tree and Logistic Regression models.

Potts et al. [60] proposed Decision Trees to classify pediatric patients into "severe" or "non-severe" Dengue cases. The stopping rules used to create the trees were described in the paper and are summarised in Table 1. Five scenarios were evaluated with different definitions of Severe Dengue: *(1)* Tree 1 considered Severe Dengue as DSS, and used four of 11 available attributes (WBC, Hematocrit (HTC), Monocyte percent (MONOP), PLT); *(2)* Tree 2 defined Severe Dengue as DHF Grade 3 or 4 or *pleural effusion index* (*PEI*) $> 15$, and used five attributes (Age, WBC, PLT, Neutrophil percent (NEUTP), Aspartate aminotransferase (AST)); *(3)* Tree 3 defined Severe Dengue as DSS or required intravenous fluid; *(4)* Tree 4 defined Severe Dengue as DSS or PLT less than 50,000; and *(5)* Tree 5 defined Severe Dengue as DSS or received fluid intervention. Trees 3, 4 and 5 were not described in the work because, according

to Potts et al. [60], they did not obtain any significant improvement in relation to Trees 1 and 2. Both Decision Trees 1 and 2 have the same initial splitting variable, WBC, reinforcing the utility of this variable in distinguishing Severe Dengue. Models were validated and tested using k-fold cross-validation ($k = 5$) and results (the work did not explicitly present any metrics for evaluating the models, such as accuracy or sensitivity. However, in the results, a table was presented with values that can be interpreted as the sensitivity of the "Severe" and "non-Severe" classes), suggested that Trees 1 and 2 achieved a sensitivity metric in excess of 90% for the "Severe" class, while the sensitivity for the "non-Severe" class was below 50%.

Phhakhounthong et al. [61] proposed a CART Decision Tree model to classify Dengue severity based on clinical and laboratory attributes. They performed a Logistic Regression analysis to determine the significance of each attribute to compose the tree. In their case, the most significant factor in predicting severe dengue was haematocrit. The results obtained from k-fold cross-validation ($k = 10$) for binary classification of Severe Dengue were 60.5% sensitivity, 65% specificity, and 64.1% accuracy. Phhakhounthong et al. [61] state that tree pruning and tuning parameters were applied to optimise the model but did not describe the settings used for the experiment. Despite having performed a feature selection with Logistic Regression, the results did not exceed 65% in the metrics evaluated.

Faisal et al. [77] sought a binary classification of Dengue risk, differentiating patients as "high risk" or "low risk". An MLP model was proposed for the classification task, and a Grid Search technique was performed to optimise the model configuration, changing four parameters: number of neurons, momentum, learning rate, and number of iterations (it is noteworthy that in the Grid Search technique process, each attribute was tested individually). Seven attributes were selected using Self Organising Map (SOM) and the model achieved 70% accuracy.

Thitiprayoonwongse et al. [51] classified a patient as DF, DHF1, DHF2 and DHF3 using a Decision Tree. Two data sets were used, one from the Srinagarindra Hospital and another from the Songklanagarind Hospital. Three experiments were performed: *(1)* using only data from the Srinagarindra Hospital; *(2)* using only Songklanagarind Hospital data; and *(3)* using data from both hospitals. The attributes of both two data sets were not described, but the attributes selected to compose the Decision Tree for each experiment were presented. In Experiment *(1)*, six attributes were selected: shock, leakage, bleeding, platelet, liver size and je-vaccine. In Experiment *(2)*, nine attributes were selected: shock, leakage, bleeding, platelet, abdominal pain, rash, uri, HTC, AST. In Experiment *(3)* eight attributes were selected: shock, leakage, bleeding, platelet, Alanine transaminase (ALT), lymp, WBC (count and minimal count). The configuration of the Decision Tree was changed only in the degree of confidence parameter in Experiments *(1)* and *(2)*; no detail was provided for Experiment *(3)*. Experiment *(1)* presented the best overall results. It is interesting that even with the addition of one more data set, Experiment *(3)* largely did not achieve superior results. The class posing the greatest classification challenge, DHF1, reported the lowest values in all three experiments although all were greater than 80%.

Arafiyah et al. [48] proposed and evaluated three models to classify DHF or not DHF—Random Forest, SVM and Naive Bayes. Unfortunately, there is no information about the model configurations. The models were trained and the metrics were calculated using the Orange tool. Random Forest achieved better results than SVM and Naive Bayes models: 79.6% accuracy, 84.1% precision, 82.2% sensitivity, 83.1% F1-Score and 89.8% AUC. No details on hyperparameter optimisation or feature selection were provided.

Fahmi et al. [52] evaluated eight models for classifying Dengue into three categories: DF, DHF and DSS. The models included NN, SVM, kNN, Decision Tree, Random Forest, Naive Bayes, AdaBoost, and Logistic Regression. The configuration of all models were described and are summarised in Table 1. Experiments were carried out in two different scenarios: *(1)*

without feature selection, and *(2)* with feature selection using the ReliefF technique (is an algorithm developed by Kira and Rendell in 1992 [95] that takes a filter-method approach to feature selection that is notably sensitive to feature interactions). In both scenarios, the best result was obtained by the NN model with 71.3% accuracy, 70.8% precision, and 71.3% sensitivity in Scenario *(1)* and with 72% accuracy, 71.5% precision, and 72% sensitivity in Scenario *(2)*. Results showed that the feature selection did provide significant improvements.

In summary, those studies in our sample addressing the diagnosis of Dengue primarily focused on binary classification (11); only two studies performed multi-class classification. Multi-class classification studies sought to classify Dengue subtypes [51] or different levels of disease severity [52]. The prevalence of binary classification reflects its simpler nature. Multi-class classification is both more complex to perform and interpret, and consequently results are often inferior to simpler models. This is reflected in our SLR [51, 52]. Tree based models (Decision Tree and derivatives) were the most common technique used in Dengue classification (10); nine of which used simple Decision Trees, often obtaining better results than other benchmark models. It is important to highlight that despite tree-based learning algorithms being broadly used for classification problems due to their simplicity for implementation and interpretability of results, the usage of imbalanced data sets can skew the performance of such models, exacerbating inadequacies inherent in the tree splitting criterion [96]. It was noted that a number of studies likely suffered model overfitting [44, 46]. Further analysis is not possible due to the lack of detail in their publications, although it should be noted that each of these studies used the smallest data sets (see Section 3.4). The usage of different data sets and lack of detail regarding both model configuration, feature selection, and hyperparameter optimisations made comparisons of these studies difficult. For example, six of the 12 studies [43, 44, 46, 48, 53, 61] did not present any description of their proposed models thus adversely impacting future reproducibility.

**3.3.2 Chikungunya.**   Only one study in the SLR sample sought to classify Chikungunya. Hossain et al. [49] proposed a Specialised Belief Rules System (BRBES) to classify Chikungunya using clinical data containing vital signs and symptoms, and considering severity classes as output (very high, high, medium and low). The BRBES system was compared with a NN, an SVM and a Fuzzy Logic Based Expert System (FLBES) as well as expert opinions. Due to the scope of our work, we consider only the ML models for analysis, i.e., NN and SVM. The NN outperformed the SVM model, obtaining an AUC of 81.1% vs 80.8%. Notwithstanding the small difference in performance between these two ML models, neither models outperformed the BRBES system. A significant limitation of this study is the lack of detail on model configurations thereby adversely impacting comparability and reproducibility.

**3.3.3 Zika.**   In relation to the Zika classification, only one study was identified. Veiga et al. [50] sought to classify suspected cases of CZS using clinical and non-clinical data. The authors compared five algorithms: kNN, CART Trees, Random Forest, AdaBoost and Gradient Boost. After performing a Grid Search, only two of the candidate models were selected and described in the final publication—*(1)* the Random Forest model was used with a data set without textual data to address a binary classification ("Discarded cases" and "Somewhat probable"); and *(2)* the Gradient Boost model was used with a data set with supplementary textual data to handle a multi-class output ("Discarded cases", "Somewhat probable", "Moderately probable" and "Highly probable"). For the binary classification, the Random Forest model obtained 91% sensitivity and and F1-Score of 83% for the "Discarded cases" class however exhibited significantly poorer results for the "Somewhat probable" class with 50% sensitivity and an F1-Score of 61%,. During the execution of Grid Search, the tree-based models obtained a similar performance and all were superior to kNN. The small but better performance of Random Forest in relation

to other tree-based models is probably due to its bootstrapping process that helps to avoid overfitting when using small data sets as per Group 1 (272 samples).

For the multi-class problem, the Gradient Boost model presented good performance mainly for the "Discarded cases" class with 91% in all metrics (precision, sensitivity and F1-score). As the amount of data used in this experiment is greater (1109 samples) than the binary classification, the Gradient Boost obtained a better performance. However, Veiga et al. [50] did not provide details on the proportion of data in each class. As such, it is not possible to analyse whether any data imbalance impacted the models performance. This is the only study where the code of the final models are available for download (https://github.com/rafael-veiga/Classification-algorithm-of-Congenital-Zika-Syndrome-characterizations-diagnosis-and-validation).

**3.3.4 Differential arboviral diagnosis.** Given the difficulties in the differential diagnosis of arboviruses discussed in Section 1, it was surprising that only one study was identified that sought to distinguish between two different arboviral diseases, in this case Dengue and Chikungunya. Lee et al. [53] proposed models to differentiate between DF, DHF and Chikungunya cases. Four experiments were presented—*(1)* DF and Chikungunya using only clinical data; *(2)* DF and Chikungunya using clinical and laboratory data; *(3)* DHF and Chikungunya using clinical data only; and *(4)* DHF and Chikungunya using clinical and laboratory data. For each classification, a Decision Tree model was developed using R software. Details on model configuration were not provided. Results suggested that Decision Trees using clinical and laboratory data outperformed the models using only clinical data.

**3.3.5 Cross-validation.** It is worth noting that a number of studies used cross-validation techniques to validate and test their models. [42, 45, 47, 50, 52, 61] used k-fold cross-validation with $k = 10$; and [60] also used k-fold but with $k = 5$. Lee et al. [53] was the only study that applied the leave-one-out (LOO) cross-validation. Commonly, cross-validations are recommended when handling with small data sets, and in an attempt to minimise the learning bias. LOO cross-validation is a type of k-fold cross-validation in which $k$ is the number of samples in the data set. Therefore, despite taking advantage of each data point, LOO cross-validation can be computationally expensive, especially if the data set is large. While Lee et al. had a data set composed of 1,034 samples, they did not mention anything about the computational effort needed to execute the experiments.

## 3.4 What attributes are considered when applying the Machine Learning and Deep Learning techniques?

**3.4.1 Summary of data sets and attributes.** Table 2 summarises the data sets used by the included studies in this SLR by number of records, number of attributes, input for models, period of the data, and location. While the number of attributes presented in this table were described as available in a focal data set, in some cases, the studies did not use all of them for training and testing their proposed models (see Sub-section 3.4.2).

Tanner et al. [42] used a data set comprising 1,200 patient records from Singapore and Vietnam with acute febrile illness. The data set is composed of 15 clinical (symptoms and vital signs) and 26 laboratory attributes. The selected attributes for the Dengue classification model were PLT, WBC and Lymphocyte count (LYMPH), body temperature, haematocrit count, and Neutrophil count (NEUT). For the classification of Dengue severity, the selected attributes were PLT, the crossover value (Ct) of the real-time Reverse transcription polymerase chain reaction (RT-PCR) for Dengue viral RNA, and the presence of anti-Dengue Immunoglobulin G (IgG) antibodies.

Fathima and Hundewale [43] used a data set comprising 5,000 records of patients with Dengue from Chennai and Tirunelveli, India. The data set includes details on 29 patient

**Table 2. Characteristics of the data sets used to evaluate Machine Learning and Deep Learning models for arboviral diseases classification.**

| Classification | Records | Attributes | Input for models | Period | Location |
|---|---|---|---|---|---|
| **Dengue** | | | | | |
| Tanner et al. [42] | 1,200 | 41 | 3 and 5 | Not described | Singapore and Vietnam |
| Fathima and Hundewale [43] | 5,000 | 29 | Not described | Not described | India |
| Sajana et al. [44] | 20 | 12 | 12 | Not described | India |
| Gambhir et al. [45] | 110 | 16 | 16 | 2015 to 2016 | India |
| Sanjudevi and Savitha [46] | 108 | 17 | Not described | Not described | Not described |
| Ho et al. [47] | 4,894 | 18 | 4, 6, 11 and 18 | 2015 | Taiwan |
| Potts et al. [60] | 1,230 | 11 | 11 | 1994–97, 1999–2002, 2004–07 | Thailand |
| Phakhounthong et al. [61] | 1,180 | 23 | 5 | 2009 to 2010 | Cambodja |
| Faisal et al. [77] | 210 | 40 | 7 | Not described | Not described |
| Thitiprayoonwongse et al. [51] | 1001 | 400 | 6, 8, and 9 | Not described | Thailand |
| Arafiyah et al. [48] | 213 | 4 | 4 | 2005 | Indonesia |
| Fahmi et al. [52] | 14,019 | 16 | 10 | 2016 to 2019 | Indonesia |
| **Chikungunya** | | | | | |
| Hossain et al. [49] | 250 | 5 | 5 | Not described | Bangladesh |
| **Zika** | | | | | |
| Veiga et al. [50] | 1,501 | 13 | 7 | 2015 | Brazil |
| **Dengue and Chikungunya** | | | | | |
| Lee et al. [53] | 1,034 | 33 | 5 | 2004 and 2008 | Singapore |

symptoms. The structure of the data set is not provided in sufficient detail to infer the extent to which the data set is balanced or unbalanced; most of the data would appear to be related to non-Dengue patients.

Sajana et al. [44] used the data collected from various medical wards of hospitals in Vijayawada, India; it comprises only 20 records with 12 attributes. As no feature selection technique is referenced in the paper, we assumed that all attributes were used for model training.

Gambhir et al. [45] used clinical and non-clinical data acquired from patients in Delhi from 2015 and 2016. The data set contains 110 records—85 positive Dengue cases and 25 negative Dengue cases. Each record has 16 attributes, of which nine are clinical data (age, gender, vomit, abdomen pain, chills, bodyache, headache, weakness, and fever) and the remainder physical examination/laboratory data (temperature, heart rate, PLT, dengue antigen NS1 or serology (Immunoglobulin M (IgM), IgG)). In the original work, these data were mistakenly classified. Gambhir et al. [45] considered age, gender, vomit, abdomen pain, chills, bodyache, headache, weakness, and fever as non-clinical data; and PLT, temperature, heart rate, dengue antigen NS1, IgM, IgG, dengue NS1 antigen as clinical.

Sanjudevi and Savitha [46] used a data set composed of 108 records with 17 attributes. The attributes were not detailed in the paper and no feature selection technique was performed.

Ho et al. [47] used data from the National University Hospital Cheng Kung (NCKUH) in Tainan City, Taiwan. The data set comprised 4,894 records of clinical and laboratory data including 2,942 cases of laboratory-confirmed Dengue cases and 1,952 non-Dengue cases. Ho et al. [47] analysed odds ratios to select four attributes and create a subset of data; two additional subsets were created with six and 11 attributes based on the initial subset. In the experiments, the data set with all attributes was also used for comparison purposes, however there was no evidence that more attributes contributed to improved model performance.

Potts et al. [60] used data from 1,384 pediatric patients with Dengue and DHF, with 11 attributes. After initial screening, 1,230 records were included in the analysis—208 cases of DHF,

374 of DF, and 648 of Other febrile illness (OFI). Data was collected in Thailand for the periods 1994 to 1997, 1999 to 2002, and 2004 to 2007.

Phakhounthong et al. [61] used a data set comprising 1,225 records related to febrile episodes in children from Angkor Hospital for Children, Cambodia. From those 1,225 records, 198 were confirmed cases of Dengue; only 38 were Severe Dengue cases. The data set included information about demographic, clinical and laboratory data. Logistic Regression was used and these five attributes were selected for model training.

Faisal et al. [77] used a data set with records of 210 patients with 40 attributes, divided into demographic, clinical, laboratory and Bioelectrical Impedance Analysis (BIA) parameters measurements to classify risk of Dengue. Laboratory data were used to classify baseline patient risk and thus create the output attribute for model training. This procedure was performed using an unsupervised model, SOM. After that, another SOM model was used to perform feature selection to define the attributes to be used as input for the proposed model; ultimately seven attributes were selected.

Thitiprayoonwongse et al. [51] used two data sets: one composed of information from 524 patients from Srinagarindra Hospital, Thailand and another with 477 patients from Songkla-nagarind Hospital, Thailand to classify DF, DHF1, DHF2 or DHF3. The Decision Tree model selected different attributes for each experiment.

Arafiyah et al. [48] used a medical data set comprising 213 records using only four clinical data sources—temperature, presence of spotting, presence of bleeding, and tornikuet test. The complete list of the attributes present in the data set was not described by the authors, so there is no way to know if or how feature selection was applied.

Fahmi et al. [52] used a data set that was provided by the Disease Prevention and Control Division in Central Java, Indonesia to classify DF, DHF or DSS. After the verification of the missing values, selection of relevant attributes, and data normalisation, the final data set comprised 14,019 records with 16 attributes including demographic, epidemiological, clinical and laboratory (hematological) information. Despite having 16 attributes available, after the application of the feature selection procedure, only ten attributes remained for model training and testing based on their importance.

As discussed, only one study addressed Chikungunya classification. Hossain et al. trained and tested their model using a data set comprising 250 records collected from various hospitals in Dhaka and Chittagong, Bangladesh. The data set had five attributes indicative of patient symptoms i.e. fever, muscle pain, joint pain, headache and swelling in the joints, each classified as high, medium or low intensity.

In relation to Zika classification, Veiga et al. [50] sought to classify suspected cases of CZS using clinical and non-clinical data. A data set with 1,501 records of live newborns suspected of microcephaly reported in the Public Health Event Registry (RESP) and the National Birth Registration System (SINASC) from Brazil was considered. This data set contains information about demographic, epidemiological, clinical (signs), laboratory (serological and others). From 13 attributes, seven were used as input for the model. Additionally, there is also textual data provided by the health professional when registering the newborns' information in the system such as reports, descriptions and other possible observations. Veiga et al. [50] separated the records into two groups where Group 1 contained only clinical and non-clinical data (272 records), and Group 2 containing clinical, non-clinical and complementary textual notes (1,109 records). The most frequent terms presented in the notes were used to assist the classification. Group 2, which considered these textual notes, obtained better results compared to Group 1.

Lee et al. [53] used demographic, epidemiological, clinical and laboratory data with 1,034 records to train and test their model to distinguish between two different arboviral diseases:

Dengue and Chikungunya. While 36 attributes are identified in the study, only five were used for training and testing the model i.e. period of symptoms, fever, fever (duration), bleeding and PLT. Of the 1,034 records, 917 were related to adult Dengue patients confirmed by Polymerase chain reaction (PCR) test, including 55 records related to DHF. 117 were records related to Chikungunya patients confirmed by RT-PCR. The Chikungunya data were collected in August 2008, while the Dengue data were collected during the large 2004 Dengue outbreak, both in Singapore.

It is important to note that none of the included studies explicitly describe or discuss how they handle imbalanced data (between-class imbalance). Given how the data sets were reported in the papers, none of the models were trained with a data set with similar number of records per class, as presented in Table 3). The data set used by Tanner et al. [42] presents this imbalancing issue in which the DSS class represents only 0.016 of the entire data set while the non -DF class presents 0.696. A similar situation is found in the data set used by Lee et al. [53] in which DHF is 0.053 and DF is 0.833. Although not explicitly mentioned, Lee at al. [53] applied the LOO cross-validation, as described in sub-section 3.3.4, that can be considered an alternative when evaluating models with imbalanced data sets. According to He et al. [96], when presented with complex imbalanced data sets, most standard learning algorithms "*fail to*

**Table 3. Distribution of samples per classes.**

|  | Classes | Samples | Proportion* |
|---|---|---|---|
| **Dengue** |  |  |  |
| Tanner et al. [42] | Non -DF | 836 | 0.696 |
|  | DF | 173 | 0.144 |
|  | DHF | 171 | 0.142 |
|  | DSS | 20 | 0.016 |
| Gambhir et al. [45] | Non-Dengue | 25 | 0.227 |
|  | Dengue | 85 | 0.772 |
| Ho et al. [47] | Non-Dengue | 1,952 | 0.398 |
|  | Dengue | 2,942 | 0.601 |
| Potts et al. [60] | OFI | 648 | 0.526 |
|  | DF | 374 | 0.304 |
|  | DHF | 208 | 0.169 |
| Phakhounthong et al. [61]** | Dengue | 160 | 0.808 |
|  | Severe Dengue | 38 | 0.191 |
| Thitiprayoonwongse et al. [51] | DF | 488 | 0.487 |
|  | DHF 1 | 222 | 0.221 |
|  | DHF 2 | 229 | 0.228 |
|  | DHF 3 | 62 | 0.061 |
| Fahmi et al. [52] | DF | 4,870 | 0.347 |
|  | DHF | 8,540 | 0.609 |
|  | DSS | 609 | 0.043 |
| **Dengue and Chikungunya** |  |  |  |
| Lee et al. [53] | DF | 862 | 0.833 |
|  | DFH | 55 | 0.053 |
|  | Chikungunya | 117 | 0.113 |

*Numbers were rounded.

**There are 982 samples of non-Dengue cases in this data set with an overall total of 1,180 records as shown in Table 2, but as this class was not considered in the problem, we did not use it for calculating the proportion.

*properly represent the distributive characteristics of the data and resultantly provide unfavorable accuracies across the classes of the data*". It happens essentially because ML models learn by reducing the error and do not take into consideration the class proportion. In health, where the minority class is commonly the positive case for the target disease (or the rare case), it is desirable that a classifier provides high accuracy for the minority class, without severely impacting on the performance of the majority class [96]. Three exceptions were identified in our sample. Gambhir et al. [45] and Ho et al. [47] used data sets in which the number of Dengue cases is larger than non-Dengue; and Fahmi et al. [52] used a large data set, in which there were more DHF cases than DF.

Additionally, the combination of imbalanced data and small sample size issue was also found in this SLR. For example, Gambhir et al. [45] used a data set composed of 110 records with 85 positive cases of Dengue and 25 negative cases. Other works also presented a small data set, such as [44, 46, 48, 49, 77], having less than 250 records each, however none of them described the distribution of the classes. In such cases, traditional learning algorithms may fail to use inductive rules over the sample space [96]. When samples are limited, the rules formed can become too specific, leading to overfitting [96]. This is likely to be the case in Sajana et al. [44] and Sanjudevi and Savitha [46]. To address these issues, some methods, such as sampling, cost-sensitivity, kernel-based and active learning [96] are available in the literature.

**3.4.2 Attributes of the data sets.** Fig 3 presents the types of attributes found in the data sets described previously. Demographic, epidemiological and clinical (symptoms, signs and co-morbidities) data were grouped as resource-limited attributes following the terminology presented by Lee et al. [53]; specific equipment is not specified for these data as they were collected at the time of the appointment. Laboratory attributes (hematological, biochemical and serological) and others are grouped as well-resourced attributes because they require specific equipment to be performed.

Table 4 presents a summary of all demographic, epidemiological and clinical data present in the data set used by the 15 included studies. Despite the focus on studies that used clinical data as input for the classifiers as per [48, 49], we also found cases in which clinical data was used together with other types of data, e.g. [42, 44, 45, 47, 50–53, 60, 61, 77]. Fathima and Hundewale [43] and Sanjudev et al. [46] neither provided details about the data set nor the attributes used in their studies. Thitiprayoonwongse et al. [51] only described the final selected attributes in the data set.

Age, gender, weight and residence (state) were the demographic information present in the data set described in [45, 47, 50, 52, 53, 60, 77]. The most common clinical data used to classify arboviral diseases were: abdominal pain, fever, temperature, and bleeding.

Table 5 presents the summary of all non-clinical (laboratory and others) data found in this SLR As expected, none of the 15 primary studies used only non-clinical data (since these works were excluded from our SLR). The non-clinical data presented in the Table 5 is quite

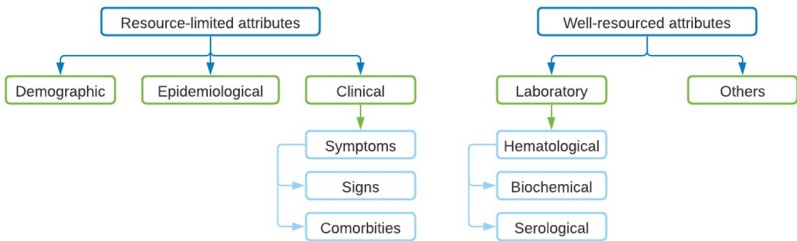

**Fig 3. Attributes found in the data sets.**

**Table 4. Summary of all demographic, epidemiological and clinical data presented in data set used by the primary studies.**

| Attributes | [42] | [43] | [44] | [45] | [46] | [47] | [60] | [61] | [77] | [51] | [48] | [52] | [49] | [50] | [53] |
|---|---|---|---|---|---|---|---|---|---|---|---|---|---|---|---|
| **Demographic data** | | | | | | | | | | | | | | | |
| Age | Ø | - | Ø | ✓★ | - | ✓★ | ✓★ | ✓ | Ø | Ø | Ø | ✓★ | Ø | ✓ | ✓ |
| Gender | Ø | - | Ø | ✓★ | - | ✓ | ✓★ | ✓ | ✓ | Ø | Ø | ✓ | Ø | ✓★ | ✓ |
| Weight | Ø | - | Ø | Ø | - | Ø | ✓ | Ø | ✓★ | Ø | Ø | Ø | Ø | Ø | Ø |
| Residence (state) | Ø | - | Ø | Ø | - | Ø | Ø | Ø | Ø | Ø | Ø | Ø | Ø | ✓ | Ø |
| **Epidemiological data** | | | | | | | | | | | | | | | |
| Period of symptoms | Ø | - | Ø | Ø | - | Ø | Ø | ✓ | Ø | Ø | Ø | ✓★ | Ø | Ø | ✓★ |
| Epidemiological week | Ø | - | Ø | Ø | - | ✓ | Ø | Ø | Ø | Ø | Ø | Ø | Ø | ✓ | Ø |
| Severity | Ø | - | Ø | Ø | - | ✓ | Ø | ✓ | Ø | Ø | Ø | Ø | Ø | ✓ | Ø |
| Japanese encephalitis vaccine | Ø | - | Ø | Ø | - | Ø | Ø | Ø | Ø | ✓★ | Ø | ✓ | Ø | Ø | Ø |
| **Clinical data** | | | | | | | | | | | | | | | |
| **Symptoms** | | | | | | | | | | | | | | | |
| Abdominal pain | Ø | - | ✓★ | ✓★ | - | Ø | ✓ | ✓ | ✓★ | ✓★ | Ø | Ø | Ø | Ø | ✓ |
| Fever | Ø | - | Ø | ✓★ | - | ✓★ | Ø | Ø | ✓ | Ø | Ø | Ø | ✓★ | Ø | ✓★ |
| Headache | ✓ | - | ✓★ | ✓★ | - | Ø | ✓ | ✓ | ✓ | Ø | Ø | Ø | ✓★ | Ø | ✓ |
| Myalgia | ✓ | - | ✓★ | ✓★ | - | Ø | Ø | Ø | ✓ | Ø | Ø | Ø | ✓★ | Ø | ✓ |
| Vomiting | ✓ | - | ✓★ | ✓★ | - | Ø | ✓ | ✓ | ✓ | Ø | Ø | Ø | Ø | Ø | ✓ |
| Arthalgya | ✓ | - | ✓★ | Ø | - | Ø | Ø | Ø | ✓ | Ø | Ø | Ø | ✓★ | Ø | Ø |
| Fever (duration) | Ø | - | Ø | Ø | - | Ø | ✓ | ✓ | Ø | ✓★ | Ø | Ø | Ø | Ø | ✓★ |
| Diarrhoea | ✓ | - | ✓★ | Ø | - | Ø | Ø | Ø | Ø | ✓ | Ø | Ø | Ø | Ø | ✓ |
| Retroorbital pain | ✓ | - | ✓★ | Ø | - | Ø | Ø | ✓ | Ø | Ø | Ø | Ø | Ø | Ø | Ø |
| Weakness | ✓ | - | Ø | ✓★ | - | Ø | Ø | Ø | ✓ | Ø | Ø | Ø | Ø | Ø | Ø |
| Chills | Ø | - | Ø | ✓★ | - | Ø | Ø | Ø | ✓ | Ø | Ø | Ø | Ø | Ø | Ø |
| Taste alteration | ✓ | - | ✓★ | Ø | - | Ø | Ø | Ø | Ø | Ø | Ø | Ø | Ø | Ø | Ø |
| Joint swelling | Ø | - | Ø | Ø | - | Ø | Ø | Ø | Ø | Ø | Ø | Ø | ✓★ | Ø | Ø |
| Anorexia | ✓ | - | Ø | Ø | - | Ø | ✓ | Ø | ✓ | Ø | Ø | Ø | Ø | Ø | ✓ |
| Nausea | ✓ | - | Ø | Ø | - | Ø | ✓ | Ø | ✓ | Ø | Ø | Ø | Ø | Ø | ✓ |
| Conjunctivitis | ✓ | - | Ø | Ø | - | Ø | Ø | Ø | ✓ | Ø | Ø | Ø | Ø | Ø | Ø |
| Cough | Ø | - | Ø | Ø | - | Ø | Ø | Ø | Ø | Ø | Ø | Ø | Ø | Ø | ✓ |
| Dizziness | Ø | - | Ø | Ø | - | Ø | Ø | Ø | ✓ | Ø | Ø | Ø | Ø | Ø | Ø |
| Itching | Ø | - | Ø | Ø | - | Ø | Ø | Ø | Ø | ✓ | Ø | Ø | Ø | Ø | Ø |
| Jaundice | Ø | - | Ø | Ø | - | Ø | Ø | Ø | Ø | ✓ | Ø | Ø | Ø | Ø | Ø |
| Sore throat | Ø | - | Ø | Ø | - | Ø | Ø | Ø | Ø | Ø | Ø | Ø | Ø | Ø | ✓ |
| Skin sensitivity | ✓ | - | Ø | Ø | - | Ø | Ø | Ø | Ø | Ø | Ø | Ø | Ø | Ø | Ø |
| **Signs** | | | | | | | | | | | | | | | |
| Temperature | ✓★ | - | ✓★ | ✓★ | - | ✓ | Ø | ✓ | Ø | ✓ | ✓★ | Ø | Ø | Ø | ✓ |
| Bleeding | ✓ | - | Ø | Ø | - | Ø | ✓ | ✓ | ✓★ | ✓★ | ✓★ | Ø | Ø | Ø | ✓★ |
| Tornikuet test | Ø | - | Ø | Ø | - | Ø | ✓★ | Ø | Ø | ✓ | ✓★ | ✓★ | Ø | Ø | Ø |
| Hepatomegaly | Ø | - | Ø | Ø | - | Ø | Ø | ✓ | ✓★ | ✓★ | Ø | ✓ | Ø | Ø | Ø |
| Shock | Ø | - | Ø | Ø | - | Ø | Ø | Ø | Ø | ✓★ | Ø | ✓★ | Ø | Ø | Ø |
| Heart rate | ✓ | - | Ø | ✓★ | - | ✓ | Ø | ✓ | Ø | ✓ | Ø | Ø | Ø | Ø | ✓ |
| Rash | ✓ | - | Ø | Ø | - | Ø | Ø | ✓ | ✓ | ✓★ | Ø | Ø | Ø | ✓[†] | ✓ |
| Pleural effusion | Ø | - | Ø | Ø | - | Ø | Ø | Ø | Ø | ✓ | Ø | ✓★ | Ø | Ø | Ø |
| Ascites | Ø | - | Ø | Ø | - | Ø | Ø | Ø | Ø | ✓ | Ø | ✓★ | Ø | Ø | Ø |
| Glasgow Coma Score | Ø | - | Ø | Ø | - | Ø | Ø | ✓★ | Ø | Ø | Ø | Ø | Ø | Ø | Ø |
| Gestacional age | Ø | - | Ø | Ø | - | Ø | Ø | Ø | Ø | Ø | Ø | Ø | Ø | ✓★ | Ø |
| Head circumference | Ø | - | Ø | Ø | - | Ø | Ø | Ø | Ø | Ø | Ø | Ø | Ø | ✓★ | Ø |

(*Continued*)

**Table 4.** (Continued)

| Attributes | [42] | [43] | [44] | [45] | [46] | [47] | [60] | [61] | [77] | [51] | [48] | [52] | [49] | [50] | [53] |
|---|---|---|---|---|---|---|---|---|---|---|---|---|---|---|---|
| Plasma leakage | ∅ | - | ∅ | ∅ | - | ∅ | ∅ | ∅ | ∅ | ✓★ | ∅ | ∅ | ∅ | ✓★ | ∅ |
| Blood Pressure | ∅ | - | ∅ | ∅ | - | ✓ | ∅ | ∅ | ∅ | ✓ | ∅ | ∅ | ∅ | ∅ | ∅ |
| Respiratory rate | ∅ | - | ∅ | ∅ | - | ✓ | ∅ | ✓ | ∅ | ✓ | ∅ | ∅ | ∅ | ∅ | ∅ |
| Flush face | ∅ | - | ∅ | ∅ | - | ∅ | ∅ | ∅ | ✓ | ∅ | ∅ | ∅ | ∅ | ∅ | ∅ |
| Palpable lymphadenopathy | ✓ | - | ∅ | ∅ | - | ∅ | ∅ | ∅ | ∅ | ✓ | ∅ | ∅ | ∅ | ∅ | ∅ |
| Birthwieght | ∅ | - | ∅ | ∅ | - | ∅ | ∅ | ∅ | ∅ | ∅ | ∅ | ∅ | ∅ | ✓ | ∅ |
| Capillary refill time | ∅ | - | ∅ | ∅ | - | ∅ | ∅ | ✓ | ∅ | ∅ | ∅ | ∅ | ∅ | ∅ | ∅ |
| **Comorbidities** | | | | | | | | | | | | | | | |
| HT | ∅ | - | ∅ | ∅ | - | ✓ | ∅ | ∅ | ∅ | ∅ | ∅ | ∅ | ∅ | ∅ | ✓ |
| NCDs (except HT) | ∅ | - | ∅ | ∅ | - | ✓ | ∅ | ∅ | ∅ | ∅ | ∅ | ∅ | ∅ | ∅ | ∅ |
| Upper respiratory infection | ∅ | - | ∅ | ∅ | - | ∅ | ∅ | ∅ | ∅ | ✓ | ∅ | ∅ | ∅ | ∅ | ∅ |

✓: data available in the data set; ∅: data not available in the data set; ★: data used as input for the models; -: data set not described;

†: maternal history of rash. Some atributes were generalized based on knowledge of the authors.

Equivalent terms: Gender [45, 47, 53, 77] or sex [48, 50, 60, 61]; Period of symptoms [52, 61] or time since onset [53]; Severity (non-hospitalized, hospitalized, Intensive Care Unit (ICU) admission and death); Myalgia [53, 77], bodyache [45, 77] or muscle pain [42, 44, 49]; Artralgya [53] or joint pain [42, 44, 49]; Fever (duration) [53, 60, 61] or days of defervescence [51]; Retroorbital pain [42, 61] or pain behind eyes [44]; Weakness [45, 61] or drowsiness [42]; Taste alteration [42] or metallic taste [44]; Anorexia [53, 60, 77] or loss of apetite [42]; Conjunctivitis [51] or red eyes [42]; Bleeding [42, 48, 51, 53, 60, 77], spotting [48], petechial rash [77], bruising [51], or hematuria [61]; Rumpel-Leed test [52], R/L test [52] or tornikuet test [48, 51, 60]; Hepatomegaly [52, 77], grown liver [51] or liver enlargement [61]; Heart rate [45, 47], pulse rate [42, 61] or tachycardia [53]; Rash [42, 51, 53, 61], macular [77], or maternal history of rash [50]; Pleural effusion [51, 52] or pleural effusion index [60]; Respiratory rate [47, 61] or dyspnea [51]; Palpable lymphadenopathy [42] or lymph node enlargement [51]; NCDs include heart disease, stroke, renal injury, severe liver disease or cancer.

diverse. The most common attribute used as input for models was the PLT used in nine studies—[42, 44, 45, 47, 51–53, 60, 61]. For model training, most non-clinical data hematological in nature e.g. PLT, WBC and HTC. Dengue IgM, Dengue IgG and Dengue NS1 antigen were used by Gamhbir et al. [45]; Zika Virus (ZIKV) RT-PCR, Toxoplasmosis, Rubella, Cytomegalovirus, Herpes Symplex, and Syphilis infections (TORCHS) serology (others except Zika) and neuroimaging reports (US, CT, MRI) were used by Veiga et al. [50]. The biochemical data used in the models were ALT, creatinine and liver size.

Lee et al. [53] compared two cases in relation to the attributes present in their data set: *(1)* a resource-limited case in which only data available at the time of hospital presentation was used (clinical data), and *(2)* a well-resourced case in which clinical and laboratory data were used for classification. As per Sub-section 3.3.4, the majority of the best results of the classification of DF, DHF or Chikungunya was obtained using a set of clinical and laboratory data. These results demonstrate that the restricted usage of clinical data for multi-classification may not be as satisfactory as when clinical and non-clinical data is combined. Based on their results, we also highlight that the use of few attributes (they considered only five attributes) is feasible for the classification of DF, DHF and Chikungunya with good performance. Regarding the number of attributes, a similar conclusion was found by Ho et al. [47]. They stated that the addition of more attributes did not provide any significant improvement in the results in any of their models, so the subset with only four attributes was able to provide as much essential information as possible and can be easily collected with minimal cost. Ho et al. [47] highlight two major findings: *(1)* their "*high-sensitivity models can be an effective surveillance tool in the pre-epidemic period*" to complement clinical diagnosis, and *(2)* high-specificity models, as in their proposal, can be exploited to identify laboratory-confirmed dengue cases at outbreak sites for real-time monitoring of epidemic trends.

**Table 5. Summary of all non-clinical data (laboratory and others) presented in the data set used by the primary studies.**

| Attributes | [42] | [43] | [44] | [45] | [46] | [47] | [60] | [61] | [77] | [51] | [48] | [52] | [49] | [50] | [53] |
|---|---|---|---|---|---|---|---|---|---|---|---|---|---|---|---|
| **Laboratory data** | | | | | | | | | | | | | | | |
| **Hematological** | | | | | | | | | | | | | | | |
| Platelet count (PLT) | ✓ ★ | - | ✓ ★ | ✓ ★ | - | ✓ ★ | ✓ ★ | ✓ ★ | ✓ | ✓ ★ | ∅ | ✓ ★ | ∅ | ∅ | ✓ ★ |
| White blood cells (WBC) | ✓ ★ | - | ✓ ★ | ∅ | - | ✓ ★ | ✓ ★ | ✓ | ∅ | ✓ ★ | ∅ | ∅ | ∅ | ∅ | ✓ ★ |
| Hematocrit (HTC) | ✓ ★ | - | ∅ | ∅ | - | ∅ | ✓ ★ | ✓ ★ | ✓ | ✓ ★ | ∅ | ✓ ★ | ∅ | ∅ | ✓ |
| Lymphocyte count (LYMPH) | ✓ ★ | - | ∅ | ∅ | - | ∅ | ✓ | ✓ | ∅ | ✓ ★ | ∅ | ∅ | ∅ | ∅ | ✓ |
| Hemoglobin (HGB) | ✓ | - | ✓ ★ | ∅ | - | ✓ | ∅ | ∅ | ∅ | ∅ | ∅ | ✓ ★ | ∅ | ∅ | ✓ |
| Neutrophil count (NEUT) | ✓ ★ | - | ∅ | ∅ | - | ∅ | ✓ | ✓ | ∅ | ∅ | ∅ | ∅ | ∅ | ∅ | ✓ |
| Lymphocyte percent (LYMPHP) | ✓ | - | ∅ | ∅ | - | ∅ | ✓ ★ | ∅ | ∅ | ∅ | ∅ | ∅ | ∅ | ∅ | ✓ |
| Neutrophil percent (NEUTP) | ✓ | - | ∅ | ∅ | - | ∅ | ✓ ★ | ∅ | ∅ | ∅ | ∅ | ∅ | ∅ | ∅ | ∅ |
| Monocyte percent (MONOP) | ✓ | - | ∅ | ∅ | - | ∅ | ✓ ★ | ∅ | ∅ | ∅ | ∅ | ∅ | ∅ | ∅ | ∅ |
| Atypical lymphocyte percent (ALYMPHP) | ∅ | - | ∅ | ∅ | - | ∅ | ✓ | ∅ | ∅ | ✓ | ∅ | ∅ | ∅ | ∅ | ✓ |
| Monocyte count (MONO) | ✓ | - | ∅ | ∅ | - | ∅ | ✓ | ∅ | ∅ | ∅ | ∅ | ∅ | ∅ | ∅ | ✓ |
| Eosinophile basofile count (EOSBAS) | ✓ | - | ∅ | ∅ | - | ∅ | ∅ | ∅ | ∅ | ∅ | ∅ | ∅ | ∅ | ∅ | ∅ |
| Eosinophile basofile percent (EOSBASP) | ✓ | - | ∅ | ∅ | - | ∅ | ∅ | ∅ | ∅ | ∅ | ∅ | ∅ | ∅ | ∅ | ∅ |
| Mean corpuscular hemoglobin (MCH) | ✓ | - | ∅ | ∅ | - | ∅ | ∅ | ∅ | ∅ | ∅ | ∅ | ∅ | ∅ | ∅ | ∅ |
| Mean corpuscular hemoglobin concentration (MCHC) | ✓ | - | ∅ | ∅ | - | ∅ | ∅ | ∅ | ∅ | ∅ | ∅ | ∅ | ∅ | ∅ | ∅ |
| MCV | ✓ | - | ∅ | ∅ | - | ∅ | ∅ | ∅ | ∅ | ∅ | ∅ | ∅ | ∅ | ∅ | ∅ |
| Mean platelet volume (MPV) | ✓ | - | ∅ | ∅ | - | ∅ | ∅ | ∅ | ∅ | ∅ | ∅ | ∅ | ∅ | ∅ | ∅ |
| Platelet distribution width (PDW) | ✓ | - | ∅ | ∅ | - | ∅ | ∅ | ∅ | ∅ | ∅ | ∅ | ∅ | ∅ | ∅ | ∅ |
| Platelet large cell ratio (PLCR) | ✓ | - | ∅ | ∅ | - | ∅ | ∅ | ∅ | ∅ | ∅ | ∅ | ∅ | ∅ | ∅ | ∅ |
| Red blood cells count (RBC) | ✓ | - | ∅ | ∅ | - | ∅ | ∅ | ∅ | ∅ | ∅ | ∅ | ∅ | ∅ | ∅ | ∅ |
| Red cell distribution width (RDW) | ✓ | - | ∅ | ∅ | - | ∅ | ∅ | ∅ | ∅ | ∅ | ∅ | ∅ | ∅ | ∅ | ∅ |
| **Biochemical** | | | | | | | | | | | | | | | |
| Alanine transaminase (ALT) | ∅ | - | ∅ | ∅ | - | ∅ | ✓ ★ | ✓ | ✓ | ✓ ★ | ∅ | ∅ | ∅ | ∅ | ✓ |
| Aspartate aminotransferase (AST) | ∅ | - | ∅ | ∅ | - | ∅ | ✓ ★ | ∅ | ✓ | ✓ ★ | ∅ | ∅ | ∅ | ∅ | ✓ |
| Creatinine | ∅ | - | ∅ | ∅ | - | ∅ | ∅ | ✓ ★ | ∅ | ∅ | ∅ | ∅ | ∅ | ∅ | ✓ |
| Albumin | ∅ | - | ∅ | ∅ | - | ∅ | ✓ | ∅ | ∅ | ✓ | ∅ | ∅ | ∅ | ∅ | ✓ |
| Protein | ∅ | - | ∅ | ∅ | - | ∅ | ∅ | ∅ | ∅ | ✓ | ∅ | ✓ | ∅ | ∅ | ✓ |
| Urea | ∅ | - | ∅ | ∅ | - | ∅ | ∅ | ✓ | ∅ | ∅ | ∅ | ∅ | ∅ | ∅ | ✓ |
| Alkaline phosphatase (ALP) | ∅ | - | ∅ | ∅ | - | ∅ | ∅ | ∅ | ∅ | ∅ | ∅ | ∅ | ∅ | ∅ | ✓ |
| Bilirubun | ∅ | - | ∅ | ∅ | - | ∅ | ∅ | ∅ | ∅ | ∅ | ∅ | ∅ | ∅ | ∅ | ✓ |
| Potassium | ∅ | - | ∅ | ∅ | - | ∅ | ∅ | ∅ | ∅ | ∅ | ∅ | ∅ | ∅ | ∅ | ✓ |
| Sodium | ∅ | - | ∅ | ∅ | - | ∅ | ∅ | ∅ | ∅ | ∅ | ∅ | ∅ | ∅ | ∅ | ✓ |
| **Serological** | | | | | | | | | | | | | | | |
| Dengue Immunoglobulin M (IgM) (ELISA) | ∅ | - | ∅ | ✓ ★ | - | ✓ | ✓ | ✓* | ∅ | ∅ | ∅ | ✓ | ∅ | ∅ | ∅ |
| Dengue Immunoglobulin G (IgG) (ELISA) | ✓ | - | ∅ | ✓ ★ | - | ∅ | ✓ | ∅ | ∅ | ∅ | ∅ | ✓ | ∅ | ∅ | ∅ |

(*Continued*)

**Table 5.** (Continued)

| Attributes | [42] | [43] | [44] | [45] | [46] | [47] | [60] | [61] | [77] | [51] | [48] | [52] | [49] | [50] | [53] |
|---|---|---|---|---|---|---|---|---|---|---|---|---|---|---|---|
| Dengue NS1 antigen (ELISA) | Ø | - | Ø | ✓★ | - | ✓ | Ø | ✓ | Ø | Ø | Ø | Ø | Ø | Ø | Ø |
| Toxoplasmosis, Rubella, Cytomegalovirus, Herpes Symplex, and Syphilis infections (TORCHS) serology† | Ø | - | Ø | Ø | - | Ø | Ø | Ø | Ø | Ø | Ø | Ø | Ø | ✓★ | Ø |
| Zika serology | Ø | - | Ø | Ø | - | Ø | Ø | Ø | Ø | Ø | Ø | Ø | Ø | ✓★ | Ø |
| Dengue antibodies (Hemagglutination-inhibition assay (HAI)) | Ø | - | Ø | Ø | - | Ø | ✓ | Ø | Ø | Ø | Ø | Ø | Ø | Ø | Ø |
| Japanese encephalitis virus (JEV) and Dengue Immunoglobulin M (IgM) (ELISA) | Ø | - | Ø | Ø | - | Ø | Ø | ✓ | Ø | Ø | Ø | Ø | Ø | Ø | Ø |
| **Molecular biology** | | | | | | | | | | | | | | | |
| Dengue Reverse transcription polymerase chain reaction (RT-PCR) | ✓ | - | Ø | Ø | - | ✓ | Ø | Ø | Ø | Ø | Ø | Ø | Ø | Ø | ✓ |
| Dengue viral load | ✓ | - | Ø | Ø | - | ✓ | Ø | Ø | Ø | Ø | Ø | Ø | Ø | Ø | Ø |
| Chikungunya Reverse transcription polymerase chain reaction (RT-PCR) | Ø | - | Ø | Ø | - | Ø | Ø | Ø | Ø | Ø | Ø | Ø | Ø | Ø | ✓ |
| **Others** | | | | | | | | | | | | | | | |
| Dengue antigen NS1 | Ø | - | Ø | ✓★ | - | ✓ | Ø | Ø | Ø | Ø | Ø | Ø | Ø | Ø | Ø |
| Urine protein | Ø | - | Ø | Ø | - | Ø | Ø | ✓★ | Ø | Ø | Ø | Ø | Ø | Ø | Ø |
| Blood in stool | Ø | - | Ø | Ø | - | Ø | Ø | ✓ | Ø | Ø | Ø | Ø | Ø | Ø | Ø |
| Dengue viral isolation | Ø | - | Ø | Ø | - | Ø | ✓ | Ø | Ø | Ø | Ø | Ø | Ø | Ø | Ø |
| Urine red blood cells | Ø | - | Ø | Ø | - | Ø | Ø | ✓ | Ø | Ø | Ø | Ø | Ø | Ø | Ø |
| **Medical imaging** | | | | | | | | | | | | | | | |
| Neuroimaging report | Ø | - | Ø | Ø | - | Ø | Ø | Ø | Ø | Ø | Ø | Ø | Ø | ✓★ | Ø |
| Chest radiography | Ø | - | Ø | Ø | - | Ø | ✓ | Ø | Ø | Ø | Ø | Ø | Ø | Ø | Ø |
| **Bioeletrical impedance** | | | | | | | | | | | | | | | |
| Extracellular Water | Ø | - | Ø | Ø | - | Ø | Ø | Ø | ✓★ | Ø | Ø | Ø | Ø | Ø | Ø |
| Body Cell Mass | Ø | - | Ø | Ø | - | Ø | Ø | Ø | ✓★ | Ø | Ø | Ø | Ø | Ø | Ø |
| Reactance | Ø | - | Ø | Ø | - | Ø | Ø | Ø | ✓★ | Ø | Ø | Ø | Ø | Ø | Ø |
| Others† | Ø | - | Ø | Ø | - | Ø | Ø | Ø | ✓ | Ø | Ø | Ø | Ø | Ø | Ø |

✓: Data available in the data set; Ø: Data not available in the data set; ★: Data used as input for the models; -: Data set not described;

†: PLT; WBC; HTC; HGB; NEUT; NEUTP LYMPH; LYMPHP; ALYMPHP; MONO; MONOP; EOSBAS; EOSBASP; MCH; MCHC; MPV; PDW; PLCR; RBC; RDW; ALT; AST; ALP; IgM; IgG; Enzyme-linked immunoassay (ELISA); HAI; JEV; RT-PCR;

*This article searched Dengue's antibodies on cerebrospinal fluid samples; TORCHS; Others features as dataset models on article [77] were resistance, phase angle, body capacitance, TRT = TBW/W, intracellular water, total body water, extracellular water, fat mass, body mass index, lean body mass, (ERB) = (ECM/BCM), basal metabolic rate, ERI = ECW/ICW. To confirm arboviruses diagnosis, some articles used WHO Dengue's criteria [51] or didn't specify how the procedure was made [44, 48, 49, 77]. Some attributes were generalised based on knowledge of the authors. Equivalent terms: PLT [42, 44, 47, 53, 60, 61, 77], maximum and minimum PLT [51], or thrombocytes [52]; WBC [42, 44, 47, 60, 61], maximum and minimal count WBC [51], or leukocyte count [53]; High and low hematocrit [51, 61], initial or diagnosis hematocrit [52], or when just one measure was made [42, 44, 45, 47, 53, 60]; RDW CV or SV [42]; Protein [53], hypoproteinemia [52] or globuline [51]; Dengue viral load [47] or Crossover threshold of Crossover threshold of Dengue Virus (DENV) RT-PCR [42].

It is interesting to note that the data sets used are quite different with regard to the number of samples and attributes. In addition, the included studies did not use similar attributes for training and testing their proposed models. In general, the included studies did not describe clinical and non-clinical data in a standardised way, making it difficult to summarise these

data without a health professional. Another challenge when analysing the data sets is related to the lack of detailed description in the included studies. Data description and experiment methodology are fundamental for replicability of studies; in half of the cases, there is no information about the model configuration. Additionally, although all studies used data sets, none of these are available for usage further adversely impacting reproducibility.

### 3.5 What are the metrics being used to evaluate the performance of the Machine Learning and Deep Learning techniques?

The common metrics used to evaluate a classifier are calculated based on a confusion matrix. The confusion matrix is a cross table that records the number of occurrences between the true classification and the classification predicted by the model [97]. It is composed of four values:

- True Positive (TP): The number of values of the principal class that the model predicts right.

- False Positive (FP): The number of values of the principal class that the model predicts wrong.

- True Negative (TN): The number of values of the secondary class that the model predicts right.

- False Negative (FN): The number of values of the secondary class that the model predicts wrong.

Fig 4 presents the metrics used to evaluate the proposed models in the literature. Some works used more than one metric and are duplicated in the graph. The evaluation metrics used by the works found in this SLR are: sensitivity, accuracy, specificity, precision, Receiver Operating Characteristic (ROC) and AUC, and F1-score. Sensitivity and accuracy were used in most studies included in the SLR sample.

**3.5.1 Sensitivity.** Sensitivity, also known as recall, was used by seven studies—[42–48, 50–53, 61]. It defines how well a model correctly predicted TP cases. It is calculated as the number of TP divided by the sum of TP and FN, as shown in Eq 1.

$$sensitivity = \frac{TP}{TP + FN} \tag{1}$$

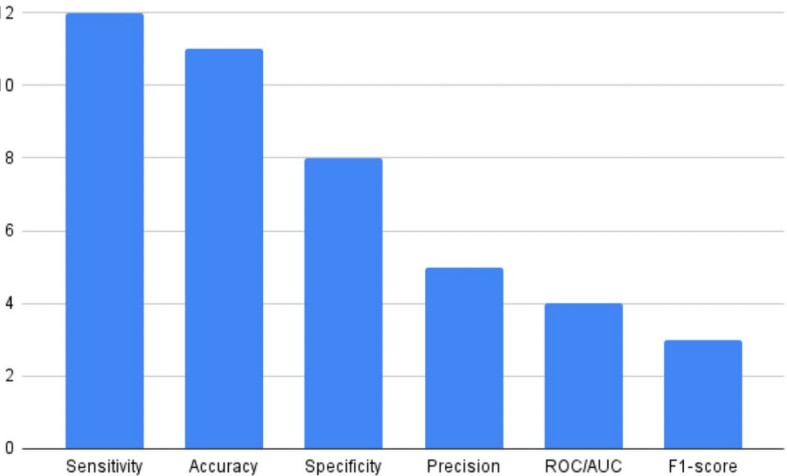

**Fig 4. Metrics used to evaluate the models proposed in the literature.**

**3.5.2 Accuracy.** Accuracy was the most common metric used among the studies in this SLR—[42–48, 50–52, 77]. It is used to find out how much a model is right. It is calculated as the sum of TP and TN divided by the total of samples, as shown in Eq 2.

$$accuracy = \frac{TP + TN}{TP + TN + FP + FN} \qquad (2)$$

**3.5.3 Specificity.** Specificity was used by five studies included in this SLR—[42, 43, 45–47, 51, 53, 61]. This metric determines how well the model correctly predicted TN cases. It is calculated by the number of TN divided by the sum of TN and FP as per Eq 3.

$$specificity = \frac{TN}{TN + FP} \qquad (3)$$

**3.5.4 Precision.** Precision was used in [44, 48, 50, 52, 61]. This metric defines how many cases classified as TP actually are TP, and is calculated as the number of TP divided by the sum of TP and FP, as shown in Eq 4.

$$precision = \frac{TP}{TP + FP} \qquad (4)$$

**3.5.5 Receiver Operating Characteristic (ROC) and Area Under the Curve (AUC).**
ROC and AUC were used in four studies—[46, 48, 49, 53]. The ROC curve is a graph to analyse the discriminating ability of the model, that is, how well the model is able to divide between two classes. It is a graph with the True Positive Rate (TPR), the sensitivity, in the $x$ axis, and the False Positive Rate (FPR), the complement of the specificity, in the $y$ axis. Based on ROC, it is possible to calculate the AUC. The AUC summarises the ROC curve in a single value, aggregating all the ROC thresholds. Its result varies between 0 and 1; an AUC of 0.5 represents a test without discriminating ability, while an AUC of 1.0 represents a test with perfect discrimination [98].

**3.5.6 F1-Score.** The F1-score is the harmonic mean between two metrics: precision and sensitivity. It is used when the objective is to seek a balance between these two metrics, being calculated as presented in Eq 5. This metric was used in [44, 48, 50].

$$F1 - score = 2 \times \frac{precision \times sensitivity}{precision + sensitivity} \qquad (5)$$

To address the imbalanced data issues mentioned earlier, more informative assessment metrics can be used to evaluate models. These include AUC, precision-recall curves and cost curves. Notwithstanding suspected imbalances, most of the included studies employed traditional metrics such as accuracy, sensitivity and specificity. Relatively few (3) used AUC [42, 46, 49].

## 4 Discussion

In this SLR on the use of ML and DL to support the clinical diagnosis of arboviral diseases, we found 963 publications, 15 of which fulfilled the inclusion criteria and were subsequently analysed in detail. We have reported our findings in five main categories: *(1)* disease focus, *(2)* ML and DL technique, *(3)* ML and DL model design, *(4)* data sets and attributes, and *(5)* evaluation metrics. Comparing the selected studies, even within these categories, due to the variation in

focal disease and region, ML and DL technique, and ML and DL model configuration, is challenging. Results are not uniformly presented. Six of the studies [43, 44, 46, 48, 53, 61] did not provide sufficient details on their proposed models, none of the selected studies provided access to their data, and only one study [50] provided details of their models online for download.

Firstly, given the low number of studies, it is important to note that there is clearly a dearth of research in the use of ML and DL to support clinical diagnosis of arboviral diseases as a whole. This paucity of research is further exacerbated when one considers that only three arboviral diseases (Dengue, Chikungunya, and Zika) feature in the selected studies, and most of the papers (12) focus on one disease, Dengue. Arboviruses, such as yellow fever, which did not feature in any of the selected studies have a significant burden. 47 countries in Africa, Central and South America have regions that are endemic for yellow fever. For example, in 2013, the burden of yellow fever in Africa alone was estimated at 84,000–170,000 severe cases and 29,000–60,000 deaths. While Dengue, Chikungunya, and Zika undoubtedly require further study, there exists a significant need for research in the wider spectrum of arboviral diseases. Given the similarity in symptoms across arboviruses, there is a surprising paucity of research on ML and DL to support differential diagnosis using clinical data. One of the most common reasons for people in low-resource areas to seek healthcare is febrile illness [99]. Such illnesses include respiratory tract infections, mononucleosis, malaria, and typhoid fever [100]). Similar to arboviruses, they typically require complex laboratory tests for confirmation, and measuring both sensitivity and specificity can be challenging. Few studies in the SLR sample pursued multi-class disease classification. Future studies could explore the efficacy of different approaches to multi-class classification for arboviruses and other febrile illnesses including *(1)* the development of multi-class algorithms, and *(2)* decomposing a multi-class problem to multiple two-class (binary) problems as per Zhou et al. [101].

Systems identified for ML and DL diagnosis of arboviruses using clinical data were generally found to be effective. However, these findings must be tempered with caution. In some cases, for example, Sanjudevi and Savitha [46] and Sajana et al. [44], the extremely high performance metrics suggest model overfitting. In both cases, there is a lack of detail on model configuration, feature selection, and hyperparameter optimisation. In the case of Sajana et al. [44], the data set is very small.

We note that only one paper made use of a DL architecture, a CNN [47], and no papers made use of ensemble methods combining DL and ML. DL models are attracting significant attention in the health domain [102], especially for dealing with unstructured data such as images, and time series data. However, our findings suggest that previous studies on the classification of arboviral diseases mostly use tabular data and are dominated by tree-based models. This is expected since traditional ML models are suitable for this type of data. To make use of tabular data with DL models, the main challenge is to reshape the data to fit it into the specific input representation. Commonly, data sets are high-dimensional and very sparse. Consequently, the challenge of reshaping a good input representation is exacerbated [103]. Another challenge for DL models working with tabular data is related to the scale and distribution of the features present in the data set. For tree-based models, this aspect is insignificant however DLs are very sensitive to this issue which can result in vanishing and exploding gradient problems [103]. Despite these challenges, one can take advantage of the powerful features that DL can inherently provide. For instance, by using a DL model and associated convolution and max pooling layers, one can reduce the time and effort involved in manual feature selection, a time-consuming task commonly used when pre-processing data sets for ML training and testing. Extant studies have proposed the transformation of tabular data to images (matrix) in order to feed them into a DL model. For example, Alvin *et al.* [104] proposed a CNN and a

LSTM to detect sepsis in neonates. Their results show that DLs outperformed selected traditional ML models, such as SVM and logistic regression, suggesting that significant improvements can be achieved with DL if data is reshaped accordingly. Future research should consider benchmarking the performance of a wider set of DL architectures, both discretely and as part of ensembles, including Recurrent Neural Networks (RNNs), Deep Belief Networks (DBNs), and Deep Stacking Networks (DSNs), amongst others, using different techniques for reshaping the data sets. In particular, the use of Long Short-Term Memory (LSTM) and Gated Recurrent Units (GRU), both types of RNNs, may prove fruitful.

While ML and DL techniques represent a significant opportunity for research and practice, they pose their own challenges. Two significant challenges are transparency and data availability. ML and DL are often referred to as *black box* models as their inner workings is too complex for a human to comprehend. As such they have been criticised for their lack of interpretability, comprehensibility, and transparency. For legal, ethical, and scientific reasons, this is a significant issue for high stakes decisions such as clinical diagnosis [105, 106]. As such, there has been numerous calls for research on explainable ML and AI (sometimes referred to as Explainable Artificial Intelligence (XAI)) [105–107] and some small but significant progress has been made in the use of XAI in clinical diagnosis, albeit not in arboviruses, the focus of this paper (see, for example, [108]).

A second significant challenge in the diagnosis of arboviral diseases using clinical data is more logistical and is related to the size and quality of available data sets. Having a sufficient data set to train and validate ML and DL models is critical. Firstly, none of the studies in the SLR sample explicitly describe or discuss imbalanced data, a common feature across the studies. Imbalanced class distribution can result in models biasing towards the majority class [109]. This problem can be addressed in a number of ways. For example, Chawla et al. [110] suggest that at the data level, one can apply random oversampling with substitution, random undersampling, directed oversampling, or oversampling with an informed generation of new samples. Similarly, Chawla et al. [110] suggest that at the algorithmic level, cost adjustments of the various classes to counteract class imbalance, adjustments in the probabilistic estimate on the tree sheet (when working with decision trees), and adjustments in the decision threshold and based on recognition rather than discrimination-based learning. Secondly, our findings suggest that most of the data sets in the selected studies were relatively small. One might argue, too small. The largest, presented in Fahmi et al. [52] comprised only 14,019 records, while the smallest comprised 20 records [44]. Thirdly, we note a potential data set shift problem. Data set shift occurs where there is a change of data source from training to testing. Consequently, there is a difference in the distributions of training and test data resulting in models learned on the training data failing on the test data [111]. Common types of dataset shift include simple covariate shift, prior probability shift, sample selection bias, and imbalanced data [112]. Each of these issues impact confidence in results and generalisability but also have practical implications for their future operational use. Overcoming some of these problems, such as sample selection bias, requires a coordinated effort by health surveillance systems and researchers worldwide and greater sharing of clinical data.

Feature selection and hyperparameter optimisation are key steps in the selection and optimisation of ML and DL models. A significant issue in many of the selected studies was the lack of detail on whether feature selection and hyperparameter optimisation were used and if so, what techniques. Only Fathima and Hundewale [43], Faisal et al. [77] and Veiga et al. [50] reported using a hyperparameter optimisation technique, grid search, to find better model configurations. Others [47, 51–53, 61, 77] explicitly reported applying feature selection techniques to find the attributes that provided the best results for their models. Eight studies provide no detail at all [42, 45–50, 60]. Assuming these studies did not use

these search and optimisation techniques, this represents an opportunity for further performance improvement.

Our review focused on studies that made use of clinical data in their ML and DL models. In some cases, this was not sole data source. For example, most of the selected studies used clinical data with other types of data, such as laboratory test results to support the diagnosis of a given arbovirus [42, 44, 45, 47, 50–53, 60, 61, 77]. Arboviral diseases are common in some of the poorest and remote regions of the world. Diagnosis based on laboratory tests requires both the availability of specialised equipment and staff to operate it. Even if available, in some instances this may add to the lapsed time and complexity of diagnosis. In contrast, a decision support tool based on clinical data using ML and DL is low-cost and rapid without the need of specialised resources.

In their guidelines for developing and reporting ML models in biomedical research, Luo et al. [113] suggest that the following evaluation metrics should be reported—sensitivity, specificity, positive predictive value, negative predictive value, AUC, and calibration plot. Our analysis suggests a significant gap between the selected studies and these guidelines. While most studies evaluated sensitivity, specificity and predictive values, few studies offered comprehensive evaluation across all metrics. For example, only four studies measured performance using ROC and AUC.

Finally, disregarding the models that seem to present overfitting or inadequate benchmarking, none of the works exceeded an accuracy of 85%. This performance may be explained by a range of factors including the quality of the training data set, and the absence or poorly executed feature selection or hyperparameter optimisation. Notwithstanding, this given the limitations and potential for improvement, we are optimistic that ML and DL offers significant opportunities for the development of low cost decision support tools to support diagnosis, particularly in remote and vulnerable areas, often characterised by poverty.

## 5 Conclusions

This SLR presented an overview of the current literature that applies ML and DL models for the classification of arboviral diseases as a support for clinical diagnosis. Of the wide range of arboviruses, ML and DL research to diagnose arboviruses based on clinical data is limited to the three most common infections—Dengue, Chikungunya and Zika. We identified the main goals were largely binary classifications. In the case of Dengue, there is evidence of more nuanced attempts at multi-class classification e.g. Dengue severity and risk. Similarly, there is some evidence of differential diagnosis both within viruses (e.g. Dengue/Severe Dengue or Zika/Congenital Zika) and between viruses however such studies are the exception. Although a limited sample, the majority of included studies focused on ML techniques rather than DL. Of the former, most were tree-based models (Decision Tree, Adaboost, Gradient Boost, Random Forest). The solitary DL model was a CNN, DenseNet. The most common evaluation metrics were accuracy, sensibility and specificity. Despite evidence of imbalanced datasets, only three included studies used AUC. In summary, ML and DL research to diagnose arboviruses is at a nascent level of maturity.

We suggest that having an efficient and comprehensive arboviral diseases clinical decision support system can improve the quality of the entire arboviral disease clinical process, thereby increasing the accuracy, precision, and throughput of diagnosis (and mitigating the risk of misdiagnosis) and associated treatment. It would also help the physicians in their decision making process and, as a consequence, improve resource utilisation and patient quality of life as a whole. However, this requires a sustained, focused, and systematic approach to research that places differential diagnosis and reproducibility at its core. This implies greater

coordination and sharing of data sets and greater detail regarding model configuration, feature selection, and hyper-parameter optimisation.

ML, and DL more specifically, have significant legal, ethical, and scientific limitations particularly with respect to healthcare decision making, not least the black box nature of many DL techniques. In terms of future research based on the results and open challenges of a SLR, we highlight the following directions, regarding the diagnosis and classification of arboviral diseases using ML and DL: *(1)* use of different data types and sources including clinical and demographic data, structured and unstructured data, for training and testing models; *(2)* applications of techniques to address imbalanced data; *(3)* greater exploration and evaluation of DL models and ensemble models, comprising ML and DL models, for arboviral classification; *(4)* greater focus on differential diagnosis within and across a wider range of arboviruses; *(5)* application of feature selection and hyperparameter optimisation techniques to fine-tune models; *(6)* consistent use of a more comprehensive set of evaluation metrics to accommodate imbalanced data, and *(7)* an easy-to-access diagnosis decision support system in remote regions allowing for intermittent connectivity.

## Acknowledgments

Authors would like to thank Conselho Nacional de Desenvolvimento Científico e Tecnológico (CNPq); Fundação de Amparo à Pesquisa do Estado do Amazonas (FAPEAM); Fundação de Vigilância em Saúde Dra. Rosemary Costa Pinto; Fundação de Amparo a Ciência e Tecnologia do Estado de Pernambuco (FACEPE); and Universidade de Pernambuco (UPE), an entity of the Government of the State of Pernambuco focused on the promotion of Teaching, Research and Extension.

## Author Contributions

**Conceptualization:** Sebastião Rogério da Silva Neto, Thomás Tabosa Oliveira, Igor Vitor Teixeira, Theo Lynn, Patricia Takako Endo.

**Data curation:** Sebastião Rogério da Silva Neto, Thomás Tabosa Oliveira, Igor Vitor Teixeira, Samuel Benjamin Aguiar de Oliveira, Theo Lynn, Patricia Takako Endo.

**Formal analysis:** Sebastião Rogério da Silva Neto, Thomás Tabosa Oliveira, Igor Vitor Teixeira, Samuel Benjamin Aguiar de Oliveira, Vanderson Souza Sampaio, Theo Lynn, Patricia Takako Endo.

**Funding acquisition:** Vanderson Souza Sampaio.

**Investigation:** Sebastião Rogério da Silva Neto, Thomás Tabosa Oliveira, Igor Vitor Teixeira, Samuel Benjamin Aguiar de Oliveira, Vanderson Souza Sampaio, Theo Lynn, Patricia Takako Endo.

**Methodology:** Sebastião Rogério da Silva Neto, Theo Lynn, Patricia Takako Endo.

**Project administration:** Patricia Takako Endo.

**Supervision:** Vanderson Souza Sampaio, Theo Lynn, Patricia Takako Endo.

**Validation:** Sebastião Rogério da Silva Neto, Theo Lynn, Patricia Takako Endo.

**Visualization:** Patricia Takako Endo.

**Writing – original draft:** Sebastião Rogério da Silva Neto, Thomás Tabosa Oliveira, Igor Vitor Teixeira, Samuel Benjamin Aguiar de Oliveira, Vanderson Souza Sampaio, Theo Lynn, Patricia Takako Endo.

**Writing – review & editing:** Sebastião Rogério da Silva Neto, Thomás Tabosa Oliveira, Igor Vitor Teixeira, Vanderson Souza Sampaio, Theo Lynn, Patricia Takako Endo.

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
