## [Decision Letter · Decision Letter 0]

13 Oct 2021

Dear Dr. Endo,

Thank you very much for submitting your manuscript "Machine learning and deep learning techniques to support the clinical diagnosis of arboviral diseases: A systematic review" for consideration at PLOS Neglected Tropical Diseases. As with all papers reviewed by the journal, your manuscript was reviewed by members of the editorial board and by two independent reviewers. In light of the reviews (below this email), we would like to invite the resubmission of a significantly-revised version that takes into account the reviewers' comments. Please take special note of the very reasonable recommendations to increase the clarity of your manuscript, including providing the rationale for the selection of the datasets, an expanded discussion of the limitations of the datasets selected and data analysis methods that were harnessed, and so forth. From an editorial perspective, these are significant considerations and issues that were laid out by the reviewers. Hopefully, the 60-day period (described below) is more than enough time for you to make these changes. This is an important and timely topic and we would like to see a thorough revision based on the critiques.

As you know, although we are optimistic about seeing a well-revised manuscript, we cannot make any decision about publication until we have seen the marked-up manuscript and your point-by-point response to the reviewers' comments (highlighting the exact changes made). Your revised manuscript is also likely to be sent to reviewers for further evaluation.

Sincerely,

Rhoel Ramos Dinglasan

Associate Editor

Stuart Blacksell

Deputy Editor

Reviewer's Responses to Questions

**Key Review Criteria Required for Acceptance?**

**Methods**

-Are the objectives of the study clearly articulated with a clear testable hypothesis stated?

-Is the study design appropriate to address the stated objectives?

-Is the population clearly described and appropriate for the hypothesis being tested?

-Is the sample size sufficient to ensure adequate power to address the hypothesis being tested?

-Were correct statistical analysis used to support conclusions?

-Are there concerns about ethical or regulatory requirements being met?

Reviewer #1: (No Response)

Reviewer #2: My main concern is why authors decide to perform the systematic review only by searching studies in one database, “Google Scholar”, and not including other databases, i.e. three significant publishers (PubMed, Springer and IEEE).

Do the authors select this database for a specific reason?. Please clarify and explain why you decide not to include more databases in the search.

**Results**

-Does the analysis presented match the analysis plan?

-Are the results clearly and completely presented?

-Are the figures (Tables, Images) of sufficient quality for clarity?

Reviewer #1: (No Response)

Reviewer #2: The authors perform exceptional work reviewing literature and explaining ML concepts to non-expert audiences interested in reading the journal (Focused on tropical infectious diseases). I suggest including the explanation of some concepts that are mentioned through the review that may help the reader to understand some points. I.e., Grid search, hyperparameters, Gamma parameters, Model overfitting.

**Conclusions**

-Are the conclusions supported by the data presented?

-Are the limitations of analysis clearly described?

-Do the authors discuss how these data can be helpful to advance our understanding of the topic under study?

-Is public health relevance addressed?

Reviewer #1: (No Response)

Reviewer #2: The authors exposed the results obtained and formulated a reflection about needs in the area and how the application of ML would be beneficial to the study and diagnosis of arbovirus diseases. Moreover, they exposed the caveats in the area and the urgent needs to address in future studies.

**Editorial and Data Presentation Modifications?**

Reviewer #1: (No Response)

Reviewer #2: Due to the nature of the review and the non-expert in ML audience expected in a journal focused on infectious diseases, I suggest integrating concepts to explain the ML methods in a more organised way, first specify that the methods discussed are supervised methods, and explain what is Deep learning.

- 3.2.3 Section (Neural Networks) authors start explaining MLP with no introduction and may seem a bit confusing.

- Some abbreviations are missing in the list (AI, XAI, MOMO%), and others are doubly explained (SLR).

**Summary and General Comments**

Reviewer #1: The authors review published studies using machine learning to predict arboviruses from clinical and laboratory data. Overall, they do a nice job of collecting the existing research, are rightfully critical of some of some of the major shortcomings of studies in this field including, overfitting, the selection of evaluation metrics and the poor documentation of code, data and methods.

Tables 3 and 4 provide a very nice summary of the features used in the studies.

In section 3.3 I was pleased to see the authors flagging studies that appear to be overfitting or inappropriately benchmarking their studies as this practice is unfortunately common in biological applications of supervised machine learning. It would be great if they explicitly flagged these studies in table 5. A discussion about the importance of regularization method like batch normalization, L1, L2 and C in SVM would have also been helpful. Many readers may not understand the importance of this for generalization.

Throughout the paper they refer to Machine learning and (ML)and Deep learning (DL) separately. DL is a subset of ML and only one study referred to DL so it would be much clearer to just refer to everything as ML except when specifically talking about the Ho et al. (2020) paper.

The review spends a lot of time discussing the use of a convolutional neural network (CNN), DenseNet, by Ho et al. (2020). They discuss CNN for a half a page on page 9 and again on page 10 and page 24. That model did not perform appreciably better than a logistic regression or random forest model. DenseNet is unique in that it connects each layer to every other layer in a feed forward neural network and essentially concatenates features rather than summing them. But like other CNN’s it relies on pooling and convolutions at different scales. While this makes sense for image data where the proximity of pixels has meaning, it is seemingly meaningless for the 16 features clinical and laboratory features placed in a random order in the Ho et al. (2020) analysis. It seems that DenseNet sort of just devolves into a multilayer perceptron or similar since the convolutions are meaningless. It would be good to point this out. More generally, I n the discussion the authors make the recommendation to invest in CNN, LSTM. The data types used in the model are not spatially or relationally structured and the amount of training data do not lend themselves well to these large models. I have concerns about this recommendation and the use of recurrent models (RNN, CNN, and LSTM) generally on this kind of data.

I appreciated that the authors highlighted the issue of imbalanced datasets in the studies reviewed. I think this could bear even more discussion. It could also be expanded to the larger issue of dataset shift problems like prior probability shift and covariance shift. I think there is likely to be major issues with the ability of these models to generalize well.

It would also be worth discussing if the studies point to general limits to the approach. Given the small number of general features it may be that 85% accuracy is about the best that can be expected. Given that this is data predominantly comes from a population where a healthcare professional suspected an arboviral infection and ordered a clinical test, a classifier with this level of accuracy may not perform better than a clinician assessing a patient with symptoms. This would suggest that it would be better to investing resources in developing low-cost clinical diagnostics with the potentially to achieve much higher accuracy.

The authors point to the lack of multiclass classification algorithms, this is a good point that could be expanded. Differential diagnostic procedures are employed by clinicians to diagnose patients, but it is not clear that any of these models considered a representative range of febrile illnesses.

Smaller items:

• P3 line 4: “weeks years” should probably read “weeks to years”

• Section 3 seems like a pretty general overview of ML classifiers more than an arbovirus specific review, perhaps this could be shortened.

• Reverse snowballing should be explained

Reviewer #2: Authors perform a systematic review to elucidate the usage of machine learning techniques in the diagnosis, classification and study of arboviral infectious diseases. The authors made an effort to explain ML techniques to non-expert readers and explore some perspectives related mainly to the usage of these technologies. The work is well presented and treat a topic of interest for the journal.

Moreover, a non-expert in ML reader would understand the article due to the precise and current representation of the scientific topic selected.

My suggestion is to accept the manuscript, and my comments are related to clarifying some concepts to give strength to the review by helping non-expert readers understand the topic.

In the conclusion section, the authors suggest future directions and open challenges to set the basis of subsequent studies applying ML to the diagnosis of arboviruses diseases.

PLOS authors have the option to publish the peer review history of their article (what does this mean?). If published, this will include your full peer review and any attached files.

Reviewer #1: Yes: Adam R. Rivers

Reviewer #2: No
---

## [Decision Letter · Decision Letter 1]

6 Dec 2021

Dear Dr. Endo,

We are pleased to inform you that your manuscript 'Machine learning and deep learning techniques to support the clinical diagnosis of arboviral diseases: A systematic review' has been provisionally accepted for publication in PLOS Neglected Tropical Diseases.

Best regards,

Rhoel Ramos Dinglasan

Associate Editor

Stuart Blacksell

Deputy Editor

Reviewer's Responses to Questions

**Key Review Criteria Required for Acceptance?**

**Methods**

-Are the objectives of the study clearly articulated with a clear testable hypothesis stated?

-Is the study design appropriate to address the stated objectives?

-Is the population clearly described and appropriate for the hypothesis being tested?

-Is the sample size sufficient to ensure adequate power to address the hypothesis being tested?

-Were correct statistical analysis used to support conclusions?

-Are there concerns about ethical or regulatory requirements being met?

Reviewer #1: (No Response)

**Results**

-Does the analysis presented match the analysis plan?

-Are the results clearly and completely presented?

-Are the figures (Tables, Images) of sufficient quality for clarity?

Reviewer #1: (No Response)

**Conclusions**

-Are the conclusions supported by the data presented?

-Are the limitations of analysis clearly described?

-Do the authors discuss how these data can be helpful to advance our understanding of the topic under study?

-Is public health relevance addressed?

Reviewer #1: (No Response)

**Editorial and Data Presentation Modifications?**

Reviewer #1: (No Response)

**Summary and General Comments**

Reviewer #1: The revision adequately addressed my minor concerns with the first draft. I have no further suggestions.

PLOS authors have the option to publish the peer review history of their article (what does this mean?). If published, this will include your full peer review and any attached files.

Reviewer #1: No

---

## [Editor Report · Acceptance letter]

30 Dec 2021

Dear Dr. Endo,

We are delighted to inform you that your manuscript, "Machine learning and deep learning techniques to support the clinical diagnosis of arboviral diseases: A systematic review," has been formally accepted for publication in PLOS Neglected Tropical Diseases.

Best regards,

Shaden Kamhawi

co-Editor-in-Chief

Paul Brindley

co-Editor-in-Chief
